# DAPO : Improving Multi-Step Reasoning Abilities of Large Language Models with Direct Advantage-Based Policy Optimization

Jiacai Liu[1,2]     Chaojie Wang[1,†]     Chris Yuhao Liu[1]     Liang Zeng[1]     Rui Yan[1]
Yiwen Sun[2]     Yang Liu[1]     Yahui Zhou[1]

[1]Skywork AI     [2]Fudan University

{jiacai.liu,chaojie.wang}@kunlun-inc.com

## Abstract

The role of reinforcement learning (RL) in enhancing the reasoning of large language models (LLMs) is becoming increasingly significant. Despite the success of RL in many scenarios, there are still many challenges in improving the reasoning of LLMs. One key challenge is the sparse reward, which introduces more training variance in policy optimization and makes it difficult to obtain a good estimation for value function in Actor-Critic (AC) methods. To address these issues, we introduce Direct Advantage-Based Policy Optimization (DAPO), a novel step-level offline RL algorithm with theoretical guarantees for enhancing the reasoning abilities of LLMs. Unlike response-level methods (such as DPO and GRPO) that the update directions of all reasoning steps are governed by the outcome reward uniformly, DAPO employs a critic function to provide step-level dense signals for policy optimization. Additionally, the actor and critic in DAPO are trained independently, ensuring that critic is a good estimation of true state value function and avoiding the co-training instability observed in standard AC methods. We train DAPO on mathematical and code problems and then evaluate its performance on multiple benchmarks. Our results show that DAPO can effectively enhance the mathematical and code capabilities on both SFT models and RL models, demonstrating the effectiveness of DAPO.

## 1 Introduction

In the rapidly evolving landscape of artificial intelligence, large language models (LLMs) have emerged as a cornerstone of natural language processing (NLP) and beyond. These models, trained on vast corpora of text data, have demonstrated an unprecedented ability to understand [16, 3], generate, and reasoning such as solving mathematical problems [61, 48, 63, 53] and code generations [23, 8]. When the token generation process of a LLM is modeled as a Markov Decision Process (MDP), it can be naturally optimized and aligned with human preference using reinforcement learning (RL) methods, known as Reinforcement Learning from Human Feedback (RLHF). Despite the success of RLHF in various fields [11, 42, 43, 55, 46, 6], it still encounters challenges and difficulties in the field of reasoning especially in long Chain-of-Thought (CoT) models. One of the key challenges is the sparsity of rewards [44, 57]. When using LLMs for mathematical problem-solving and code generation, rewards are only assigned to the terminal tokens. This implies that the intermediate tokens receives no direct reward, and the optimization direction relies solely on the backpropagation of the reward from the terminal token. Consequently, there are two issues in practical RL training:

---

† Corresponding author.

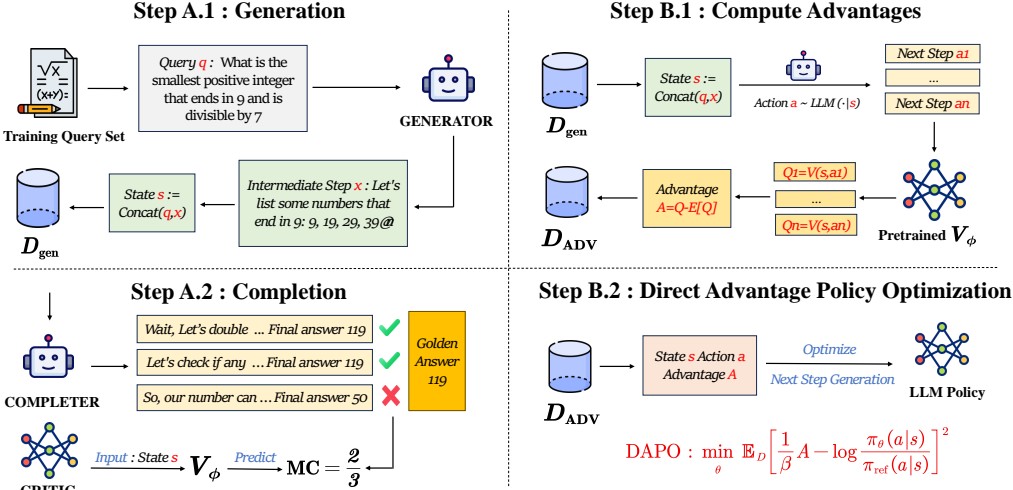

Figure 1: **Direct Advantage-Based Policy Optimization (DAPO).** The whole training procedure of DAPO consists of two individual stages : 1) **Critic Training (left)**. Given the training query set $\mathcal{Q}$, for each query $q \in \mathcal{Q}$, DAPO uses the generator $\pi_{\text{gen}}$ to generate multiple rollouts from $q$ and derive a dataset of training states denoted as $\mathcal{D}_{\text{gen}}$. For each state in $s \in \mathcal{D}_{\text{gen}}$, DAPO uses the completer (often $\pi_{\text{ref}}$) to sample multiple sub-trajectory from $s$ and collects the MC value as the estimation of the true value. Then a critic network $V_\phi$ is trained to approximate the value function of the completer. 2) **Policy Optimization (right)**. After extracting multiple next steps $\{a_i\}_{i=1}^{n}$ of each state $s$ from the completions in step A.1, DAPO then uses the trained critic to compute the advantages for all actions by $\forall i \in [n] : A_i = Q_i - \frac{1}{n}\sum_{j=1}^{n} Q_j$ and $Q_i$ is the predicted value of Concat$(s, a_i)$. Finally, DAPO fits the policy ratio to the advantage to optimize the generation of reasoning steps.

**High training variance induced by response-level algorithms.** Response-level algorithms (such as DPO [42], GRPO [48], ReMax[30] and etc) treat the complete response as a unified whole and the gradient directions across all reasoning steps within the response are governed by the final reward uniformly. Although response-level algorithms have recently achieved great success, such as Deepseek-R1 [10], the sparsity of the reward indeed introduces more training variance and addressing the training variance introduced by it would lead to a larger performance improvement. See Section D for detailed discussions.

**Bad value estimation in actor-critic methods.** When applying standard actor-critic methods [46, 15, 12] to LLMs, due to the sparsity of the reward and vast generation policy, the critic often fails not only to learn a good semantic representation but also to provide the fine-grained credit assignment for policy optimization, leading to the poor performance of online actor-critic methods [25, 62] such as PPO [46]. Besides, these methods train the actor and critic simultaneously and the regression targets of both actor and critic are non-stationary and change in response to the updates of the other during the training process. This interwoven update mechanism also leads to an unstable training process that is prone to collapse [27, 12] especially when the critic is not a good approximation of the true value function.

In order to overcome these challenges, we introduce Direct Advantage-Based Policy Optimization (DAPO), a new step-level offline actor-critic method designed to enhance LLM's performance in reasoning tasks. In order to reduce the training variance induced by the sparsity of the reward, we use a critic function to provide dense training signals. To ensure that the critic function is well-trained, DAPO trains the critic individually before policy optimization to ensure it is a good approximation of the true state value function. The procedure of DAPO is visualized in Figure 1. We summarize our contributions as follows:

- We introduce DAPO, a new step-level, offline RL method that learns from all generated samples. DAPO optimizes the generation of the reasoning steps by sampling multiple candidate steps for each intermediate step $s$ and updating the policy according to the

advantages of each candidate step. Our main theoretical result (see Theorem 3.2) shows that DAPO will produce a better policy until the reference policy is already optimal.

- We conduct extensive experiments to demonstrate the effectiveness of DAPO. DAPO improves the performance on multiple mathematical and coding benchmarks. Our results empirically shows that DAPO consistently outperforms both the response-level baseline GRPO and the actor-critic baseline PPO on multiple base models .

## 2 Preliminaries

In this section, we introduce the mathematical formulation of RL problem we studied and some important related theoretical results.

**RL reasoning.** The objective for RL reasoning can be formulated as

$$\max_{\pi} \mathbb{E}_{x \sim \mu} \left[ \mathbb{E}_{y \sim \pi(\cdot|x)} \left[ r(x, y) \right] - \beta \mathrm{KL} \left( \pi_x, \pi_x^{\mathrm{ref}} \right) \right], \tag{1}$$

where $\mu$ is the distribution of training prompt $x$, $y$ is the response sampled from policy $\pi(\cdot|x)$, $r(x, y) \in \{0, 1\}$ is the binary reward for the correctness of the response, $\pi_x, \pi_x^{\mathrm{ref}}$ are short for $\pi(\cdot|x), \pi_{\mathrm{ref}}(\cdot|x)$ respectively, $\pi_{\mathrm{ref}}$ is the reference policy, $\beta \geq 0$ is the coefficient of KL regularization.

**Equivalency to multi-step decision problem.** We use the line break \n as the delimiter for reasoning steps. By doing so, a response $y$ can be divided into multiple reasoning steps, i.e. $y = (a_0, ..., a_{T-1})$, where $\forall t \geq 0 \, a_t$ is the reasoning step and $T$ is the total number of steps. Thus one can easily show that the objective 1 is equivalent to following step-level decision problem:

$$\max_{\pi} \mathbb{E}_{x \sim \mu} \left[ \mathbb{E}_{\forall t: a_t \sim \pi(\cdot|s_t)} \left[ \sum_{t=0}^{T-1} r(s_t, a_t) - \beta \mathrm{KL} \left( \pi_{s_t}, \pi_{s_t}^{\mathrm{ref}} \right) | s_0 = x \right] \right], \tag{2}$$

where $s_t := \mathrm{Concat}(x, a_0, ..., a_{t-1})$ is the state (context prefix), the reasoning step $a_t$ is sampled from policy $\pi(\cdot|s_t)$ (here we use $\pi$ to denote both step generation and response generation policy with a slightly abuse of the notation), $\pi_{s_t}, \pi_{s_t}^{\mathrm{ref}}$ are short for $\pi(\cdot|s_t), \pi_{\mathrm{ref}}(\cdot|s_t)$ respectively and the reward function $r$ satisfies

$$r(s_t, a_t) = \begin{cases} r(x, y) & t = T - 1 \\ 0 & t < T - 1. \end{cases}$$

For any state $s \in \mathcal{S}$, we define as

$$V_{\beta}^{\pi}(s) := \mathbb{E}_{\forall t: a_t \sim \pi(\cdot|s_t)} \left[ \sum_{t=0}^{T-1} r(s_t, a_t) - \beta \mathrm{KL} \left( \pi_{s_t}, \pi_{s_t}^{\mathrm{ref}} \right) | s_0 = s \right]. \tag{3}$$

Thus the objective (2) is equivalent to finding a policy $\pi$ that maximizes the state value function, i.e.,

$$\max_{\pi} \mathbb{E}_{x \sim \mu} \left[ V_{\beta}^{\pi}(x) \right]. \tag{4}$$

We denote $\pi_{\beta}^*$ as the optimal policy of objective (4) and $V_{\beta}^* := V_{\beta}^{\pi_{\beta}^*}$ as the state value function of optimal policy. We define the state space $\mathcal{S}$ as the set of all possible states that can be encountered in the response $y$ and the action space $\mathcal{A}$ as the set of all possible singular reasoning steps. For simplicity of analysis, we introduce the following assumptions in this work.

**Assumption 2.1.** We assume there exists a constant $H_1 < \infty$ such that for any policy $\pi$ and initial state $s_0 \in \mathcal{S}$, the random decision horizon T satisfies $T < H_1$ almost surely. We also assume that there exists a constant $H_2 < \infty$ such that for any singular reasoning step $a \in \mathcal{A}$, the token length of $a$ is no greater than $H_2$ so that the state space $\mathcal{S}$ and action space $\mathcal{A}$ are both finite.

**More value functions.** We define the KL-constrained state-action value function as

$$Q_{\beta}^{\pi}(s, a) := r(s, a) + V_{\beta}^{\pi}(s \circ a), \tag{5}$$

where $s \circ a := \mathrm{Concat}(s, a)$. The advantage function $A_{\beta}^{\pi}$ is defined as

$$A_{\beta}^{\pi}(s, a) := Q_{\beta}^{\pi}(s, a) - V_{\beta}^{\pi}(s) - \beta \log \frac{\pi(a|s)}{\pi_{\mathrm{ref}}(a|s)}. \tag{6}$$

We denote $Q_{\beta}^*(s, a) := r(s, a) + V_{\beta}^*(s)$. For unregularized state value functions, i.e. $\beta = 0$, we use $V^{\pi}, Q^{\pi}, A^{\pi}$ instead of $V_0^{\pi}, Q_0^{\pi}, A_0^{\pi}$ to be aligned with the standard RL literatures.

**Bellman operators** For arbitrary policy $\pi \in \Pi$ and any function $V : \mathcal{S} \to \mathbb{R}$, the Bellman operator $\mathcal{T}_\beta^\pi$ is defined as

$$\forall s \in \mathcal{S} : [\mathcal{T}_\beta^\pi V] (s) = \mathbb{E}_{a \sim \pi(\cdot|s)} \left[ r(s,a) + V(f(s,a)) - \beta \log \frac{\pi(a|s)}{\pi_{\text{ref}}(a|s)} \right]. \tag{7}$$

**Performance difference lemma.** Following is the key theoretical analysis tool used in this work.

**Lemma 2.2.** *For arbitrary two policies $\pi, \tilde{\pi} \in \Pi$ and any distribution $\rho \in \Delta(\mathcal{S})$,*

$$V_\beta^\pi (\rho) - V_\beta^{\tilde{\pi}} (\rho) = \mathbb{E}_{s \sim d_\rho^\pi} \left[ \mathcal{T}_\beta^\pi V_\beta^{\tilde{\pi}} (s) - V_\beta^{\tilde{\pi}} (s) \right], \tag{8}$$

*where $V_\beta^\pi (\rho) := \mathbb{E}_{s \sim \rho} \left[ V_\beta^\pi (s) \right]$ and the visitation measure $d_\rho^\pi$ is defined as,*

$$d_\rho^\pi (s) := \mathbb{E}_{\forall t : a_t \sim \pi(\cdot|s_t)} \left[ \sum_{t=0}^{T-1} 1 \{s_t = s\} | s_0 \sim \rho \right].$$

The proof of this lemma is deferred to the Appendix C. We also put some additional theoretical results about MDP on Appendix B.

**Remark 2.1.** *Note that our definition of visitation measure, $d_\rho^\pi$ is not standard as the one in [1, 33]. The reason is that the planing horizon in problem (2), i.e. $T$, is random while it is the constant $\frac{1}{1-\gamma}$ in [1, 33]. Thus we do not try to normalize $d_\rho^\pi$ to a probability distribution by dividing $T$ directly.*

## 3 Direct Advantage-Based Policy Optimization

In this section, we present the details of DAPO. We only consider the case where a binary reward $r(s,a) \in \{0,1\}$ is assigned in the terminal state since it is the most common situation in the LLMs setting. We first present the objective of policy optimization with related theoretical results in Section 3.1 assuming that one has access to the true value function.

### 3.1 Policy objective

We first derive the policy optimization objective in one single iteration provided with the reference policy $\pi_{\text{ref}}$. In the perspective of policy optimization, the most important quantity may be the advantage function. Recall Lemma 2.2. The performance difference for any policy $\pi \in \Pi$ w.r.t $\pi_{\text{ref}}$ on arbitrary $\rho \in \Delta(\mathcal{S})$ is measured as

$$V_\beta^\pi (\rho) - V_\beta^{\pi_{\text{ref}}} (\rho) := \mathbb{E}_{s \sim d_\rho^\pi} \left[ \underbrace{\mathcal{T}_\beta^\pi V_\beta^{\pi_{\text{ref}}} (s) - V_\beta^{\pi_{\text{ref}}} (s)}_{:= \mathcal{I}_s (\pi, \pi_{\text{ref}})} \right],$$

where $\mathcal{I}_s (\pi, \pi_{\text{ref}})$ can be regarded as the policy improvement at state $s$. By the definition of Bellman operator $\mathcal{T}_\beta^\pi$ in (7) and that of function $V_\beta^\pi$ in (3), one can easily show that

$$\mathcal{I}_s (\pi, \pi_{\text{ref}}) = \mathbb{E}_{a \sim \pi(\cdot|s)} \left[ A^{\pi_{\text{ref}}} (s,a) - \beta \log \frac{\pi(a|s)}{\pi_{\text{ref}}(a|s)} \right].$$

A natural idea that ensures $V_\beta^\pi (\rho) - V_\beta^{\pi_{\text{ref}}} (\rho) \geq 0$ might be to increase $\mathcal{I}_s (\pi, \pi_{\text{ref}})$ on as many states $s$ as possible. For ease of understanding, we first focus on how to increase $\mathcal{I}_s (\pi, \pi_{\text{ref}})$ at an arbitrary state $s \in \mathcal{S}$. Considering parameterized policy $\pi_\theta$, the most directly and easily method may be to do multi gradient ascent steps w.r.t $\mathcal{I}_s (\pi_\theta, \pi_{\text{ref}})$, i.e.,

$$\forall K \in \mathbb{N}^+, \theta_K = \theta_0 + \eta \sum_{k=0}^{K-1} \nabla_\theta \mathcal{I}_\theta (\pi_{\theta_k}, \pi_{\text{ref}}), \tag{9}$$

where $\eta$ is the step size. Following lemma gives an equivalent form of $\nabla_\theta \mathcal{I}_s (\pi_{\theta_k}, \pi_{\text{ref}})$.

**Lemma 3.1.** *If we define the surrogate function*

$$\mathcal{H}_s^k(\theta) := \frac{1}{2}\mathbb{E}_{a\sim\pi_{\theta_k}(\cdot|s)}\left(\frac{1}{\beta}A^{\pi_{\text{ref}}}(s,a) - \log\frac{\pi_\theta(a|s)}{\pi_{\text{ref}}(a|s)}\right)^2,$$

*then* $\nabla_\theta\mathcal{I}_s(\pi_{\theta_k},\pi_{\text{ref}}) = -\beta\nabla_\theta\mathcal{H}_s^k(\theta)$.

By Lemma 3.1, the update (9) is equivalent to

$$\theta_K = \theta_0 - \eta\beta\sum_{k=0}^{K-1}\nabla_\theta\mathcal{H}_s^k(\theta_k). \tag{10}$$

We now to turn to the offline policy optimization problem setting for the sake of implementation simplicity and data efficiency. A natural question is that can we implement (10) approximately in an offline dataset without sampling the on-policy rollouts at each iteration? Our answer is affirmative. We find that (which will be proved latter), just solving the problem

$$\min_\theta \frac{1}{2}\mathbb{E}_{a\sim\nu(\cdot|s)}\left(\frac{1}{\beta}A^{\pi_{\text{ref}}}(s,a) - \log\frac{\pi_\theta(a|s)}{\pi_{\text{ref}}(a|s)}\right)^2, \tag{11}$$

where $\nu \in \Delta(\mathcal{A})$ can be any exploratory sampling distribution, can still yield a policy that $\mathcal{I}_s(\pi_\theta,\pi_{\text{ref}}) > 0$ unless $\pi_{\text{ref}} = \pi_\beta^*$. Taking the consideration of optimizing multiple states simultaneously (weighted by a state sampling distribution $\nu_\mathcal{S}$), the objective of DAPO is

$$\min_\theta \mathcal{L}(\theta) := \frac{1}{2}\mathbb{E}_{s\sim\nu_\mathcal{S},a\sim\nu_\mathcal{A}(\cdot|s)}\left[\left(\frac{1}{\beta}A^{\pi_{\text{ref}}}(s,a) - \log\frac{\pi_\theta(a|s)}{\pi_{\text{ref}}(a|s)}\right)^2\right] \tag{12}$$

**Remark 3.1.** *We give a detailed discussion in Section D on how DAPO is specifically designed to improve multi-step reasoning abilities of LLMs, thereby distinguishing it from response-level optimization algorithms such as DPO [42], GRPO [48] and etc.*

Notice that the state space $\mathcal{S}$ and the action space $\mathcal{A}$ are both finite under the assumption 2.1. Following theorem establish the validity of Loss (12) and shows that the solution of (12) is a better policy w.r.t $\pi_{\text{ref}}$ on objective (4).

**Theorem 3.2** (Monotonic improvement). *Suppose $\forall s \in \mathcal{S}, a \in \mathcal{A}, \nu_\mathcal{S}(s) > 0, \nu_\mathcal{A}(a|s) > 0, \pi_{\text{ref}}(a|s) > 0$. Let $\pi^+$ be the solution in (12). Then for any state $s \in \mathcal{S}$, there exists a function $\lambda_s : \Delta(\mathcal{A}) \to [0, +\infty)$ such that*

$$V_\beta^{\pi^+}(\mu) - V_\beta^{\pi_{\text{ref}}}(\mu) \geq \mathbb{E}_{s\sim d_\mathcal{D}^{\pi^+}}[\lambda_s(\nu_\mathcal{A})] \geq 0.$$

*The equality holds if and only if $\pi_{ref} = \pi_\beta^*$.*

**Remark 3.2.** *The function $\lambda_s$ serves as a lower bound of the policy improvement $\mathcal{I}_s(\pi^+, \pi_{\text{ref}})$ in each states $s$. Since Theorem 3.2 is established for arbitrary exploratory $\nu_\mathcal{A}$, $\lambda_s$ should be a function of $\nu_\mathcal{A}$. For the case $\nu_\mathcal{A}(\cdot|s) = \pi_{\text{ref}}(\cdot|s)$, one can show that*

$$\lambda_s(\pi_{\text{ref}}) = \beta \cdot \text{KL}(\pi_{\text{ref}}(\cdot|s), \pi^+(\cdot|s)),$$

*by combining (20) and (21) in the proof of Theorem 3.2. It remains as a great interest for us to derive a more precise expression of $\lambda_s$ and figure out which $\nu_\mathcal{A}$ should yield biggest lower bound.*

Theorem 3.2 implies that DAPO can improve the performance until there is no room for improvement in the trust region of $\pi_{\text{ref}}$. Theorem 3.2 also establishes the monotonic improvement property on unregularized state value function. This can be verified by

$$V^{\pi^+}(\mu) = V_\beta^{\pi^+}(\mu) + \mathbb{E}_{s\sim d_\mu^{\pi^+}}\left[\text{KL}(\pi^+(\cdot|s),\pi_{\text{ref}}(\cdot|s))\right] \geq V_\beta^{\pi^+}(\mu) \geq V_\beta^{\pi_{\text{ref}}}(\mu) = V^{\pi_{\text{ref}}}(\mu).$$

**Implementation of DAPO.** In practice, DAPO can be implemented by minimizing (12) on an offline training dataset $\mathcal{D} = \left\{\left(s_i, a_i, \hat{A}_i\right)\right\}_{i=1}^N$, where $s_i \sim \nu_\mathcal{S}$, $a_i \sim \nu_\mathcal{A}(\cdot|s)$ and $\hat{A}_i$ is an estimation of true advantage $A^{\pi_{\text{ref}}}(s_i, a_i)$. The whole procedure can be found in Algorithm 1.

**Advantage estimation.**   Recall the definition of Q function in (5) and advantage function in (6). For non-terminal states, one has $r(s, a) = 0$ and

$$A^{\pi_{\mathrm{ref}}}(s, a) = Q^{\pi_{\mathrm{ref}}}(s, a) - V^{\pi_{\mathrm{ref}}}(s) = V^{\pi_{\mathrm{ref}}}(s \circ a) - \mathbb{E}_{a' \sim \pi_{\mathrm{ref}}(\cdot | s)}\left[V^{\pi_{\mathrm{ref}}}(s \circ a')\right].$$

Thus in our implementation, we sample multiple actions $\{a_i\}_{i=1}^M$ for each state $s$ from $\pi_{\mathrm{ref}}(\cdot | s)$ and estimate the advantage of each action by

$$\forall i \in [m] : \hat{A}(s, a_i) := V_\phi(s \circ a_i) - \frac{1}{M}\sum_{j=1}^M V_\phi(s \circ a_j). \tag{13}$$

Here $V_\phi \approx V^{\pi_{\mathrm{ref}}}$ is a pretrained critic function. Since (13) is fully determined by the critic $V_\phi$, thus the training of $V_\phi$ plays a crucial role for the optimization of training states. In the following, we introduce the optimization method of $V_\phi$.

## 3.2   Critic objective

Generally speaking, the critic optimization problem for any target policy $\pi$ can be formulated as

$$\min_\phi \mathbb{E}_{s \sim \mathcal{D}}\left[\mathcal{L}\left(V_\phi(s), V^\pi(s)\right)\right], \tag{14}$$

here $\mathcal{L}$ is a sample-wise loss function, $\mathcal{D}$ is the distribution of training states. However, since the target $V^{\pi_{\mathrm{ref}}}(s)$ can not be accessed to in advance, we need to construct the estimate of $V^\pi$ as the training samples for the optimization of $V_\phi$. One can show that

$$V^\pi(s) = \mathbb{E}\left[\sum_{t=0}^{T-1} r(s_t, a_t) | s_0 = s\right] = \mathbb{P}^\pi\left(r(s_{T-1}, a_{T-1}) = 1 | s_0 = s\right).$$

Thus, one can firstly use a behavior policy $\pi_{\mathrm{gen}}$ (also called generator) to generate a set of training states $s$. Then for each training state $s$, one can use $\pi$ as the completer to sample $N$ sub-trajectories from $s$, $\left\{\tau_i = \left(s, a_0^{(i)}, a_1^{(i)}, ..., a_{T_i-1}^{(i)}\right)\right\}_{i=1}^N$, and construct the empirical MC mean,

$$\mathrm{MC}_N(s) := \frac{1}{N}\sum_{i=1}^N r\left(s_{T_i-1}, a_{T_i-1}\right),$$

as the estimation of $V^\pi(s)$, which is guaranteed to converge to $V^\pi(s)$ as $n \to \infty$ by the strong law of large numbers. Notice that the state value $V^\pi(s)$ is a probability. Thus we the binary cross-entropy loss in our implementation, i.e.,

$$\mathcal{L}_{\mathrm{BCE}}(y, y_{\mathrm{pred}}) = -\left(y\log(y_{\mathrm{pred}}) + (1 - y)\log(1 - y_{\mathrm{pred}})\right)$$

as the sample-wise loss function $\mathcal{L}$ in (14) to avoid gradient diminishing if a MSE loss is applied (which is more common in RL literatures). Finally, the critic optimization objective is

$$\min_\phi \mathbb{E}_{s \sim \mathcal{D}}\left[\mathcal{L}_{\mathrm{BCE}}\left(\mathrm{MC}_N(s), V_\phi(s)\right)\right]. \tag{15}$$

**Remark 3.3** (Should we treat $V_\phi$ as a PRM?). *Its worthy to note that the critic training method we presented above is mostly aligned with [56]. However, we argue that the $V_\phi$ should be treated as a value function other than a process reward model. Here are the reasons : 1) From the optimization perspective, $V_\phi$ is trained to approximate the true state value of the completer. 2) From the RL perspective, using $V_\phi$ as a reward function will deviate from the original RLHF optimization objective and easily lead to a serve reward hacking issue on the number of reasoning steps as shown in [13].*

## 4   Experiments

In this section, we conduct DAPO on mathematical and coding datasets individually. The experimental setup is detailed in Section 4.1, and specific conclusions and benchmark results of our experiments is presented in Section 4.2.

| Model | In Domain | Out of Domain | | | |
|---|---|---|---|---|---|
| | MATH | GSM8K | Minerva MATH | Olympiad Bench | College Math |
| **SFT Models** | | | | | |
| Skywork-Math-Llama | 41.90 | 61.49 | 5.51 | 18.68 | 24.85 |
| + DAPO | $46.88^{+4.98}$ | $67.55^{+6.06}$ | $7.34^{+1.82}$ | $22.38^{+3.70}$ | $25.87^{+1.03}$ |
| Llama-3.1-8B-Instruct | 49.42 | 85.29 | 26.48 | 16.14 | 30.91 |
| + DAPO | $53.62^{+4.20}$ | $86.74^{+1.45}$ | $23.54^{-2.94}$ | $20.01^{+3.88}$ | $30.91^{+1.63}$ |
| OpenO1-Llama-8B-v0.1 | 52.73 | 85.99 | 29.04 | 19.86 | 29.10 |
| + DAPO | $60.33^{+7.60}$ | $88.77^{+2.78}$ | $29.42^{+0.39}$ | $24.44^{+4.59}$ | $32.12^{+3.02}$ |
| Qwen2.5-72B-instruct | 82.90 | 95.40 | 46.30 | 45.45 | 43.00 |
| + DAPO | $84.70^{+1.80}$ | $95.70^{+0.30}$ | $50.00^{+3.70}$ | $47.70^{+2.25}$ | $43.40^{+0.40}$ |
| **RL Models** | | | | | |
| Qwen2-Math-7B-Instruct | 74.46 | 89.38 | 40.07 | 34.36 | 41.59 |
| + DAPO | $75.41^{+0.95}$ | $89.45^{+0.07}$ | $37.51^{-2.56}$ | $37.03^{+2.67}$ | $42.23^{+0.64}$ |
| Skywork-O1-Open-Llama3.1-8B | 78.10 | 91.64 | 26.10 | 43.11 | 40.40 |
| + DAPO | $79.81^{+1.71}$ | $92.81^{+1.17}$ | $29.77^{+3.67}$ | $44.76^{+1.65}$ | $40.26^{-0.14}$ |
| Qwen2.5-Math-7B-Instruct | 83.42 | 95.78 | 40.06 | 38.96 | 42.65 |
| + DAPO | $84.86^{+1.44}$ | $96.14^{+0.36}$ | $41.56^{+1.50}$ | $41.25^{+2.29}$ | $41.97^{-0.67}$ |

Table 1: Performance of models optimized via DAPO on mathematical benchmarks. We use zero-shot prompting and greedy decoding for all evaluations in the table. Across a wide range of model families and out-of-domain math word problem benchmarks, DAPO achieves steady improvement with only 7.5K training samples from MATH [18].

| SFT Model | Out of Domain | | | | |
|---|---|---|---|---|---|
| | HumanEval | HumanEval+ | MBPP | MBPP+ | LiveCodeBench |
| Llama-3.1-8B-Instruct | 72.0 | 66.5 | 72.0 | 56.9 | 18.8 |
| + DAPO | $75.0^{+3.0}$ | $68.9^{+2.4}$ | $77.0^{+5.0}$ | $66.1^{+9.2}$ | $20.9^{+2.1}$ |
| OpenO1-Llama-8B-v0.1 | 69.5 | 61.0 | 69.8 | 58.7 | 16.1 |
| + DAPO | $72.0^{+2.5}$ | $64.6^{+3.6}$ | $75.9^{+6.1}$ | $63.8^{+5.1}$ | $13.7^{-2.4}$ |

Table 2: Performance of models optimized via DAPO on code benchmarks. We use zero-shot prompting and greedy decoding for all evaluations in the table. DAPO consistently improves performance when starting with both regular and reasoning LLMs, even when only 4K training samples from TACO [29] is used.

## 4.1 Experimental setup

**Training dataset.** For all the DAPO experiments on mathematics, we utilize only the 7500 training problems from the dataset MATH [18] to generate the advantage datasets for DAPO training, requiring no additional human annotations. In particular, we only use the questions and the corresponding golden answers in the dataset while the provided solutions are not used for training. For coding experiments, For coding experiments, we subsample the TACO [29] dataset to compile approximately 4,000 competition-level programming questions derived from real-world scenarios. We utilize its provided unit test cases to evaluate whether a solution is accurate.

**Base models.** Considering the reproducibility and of DAPO, our experiments are taken over of several open-source language models including general and math-specific models. For general models, we consider Llama-3.1-8B-Instruct [52] , OpenO1-Llama-8B-v0.1 [39], Skywork-O1-Open-Llama3.1-8B [40], Qwen2.5-72B-Instruct [51]. For math-specific models, we consider Skywork-Math-Llama [63], Qwen2-Math-7B-Instruct [59] and Qwen2.5-Math-7B-Instruct [60]. Notice that Skywork-O1-Open-Llama3.1-8B, Qwen2-Math-7B-Instruct and Qwen2.5-Math-7B-Instruct are already trained by RL algorithms [48]. Thus we conduct continue-RL training on three models by DAPO.

**Benchmarks & Metrics.** For mathematical experiments, we evaluate performance on English mathematical benchmarks. In addition to the 5000 test problems from MATH [18], We also add 4 out-of-domain benchmarks, GSM8K [9], Minerva Math [28], Olympiad Bench [17] and College Math [50], to test the performance generalization of DAPO. For coding experiments, we evaluate DAPO on several widely-used benchmarks in code generation, i.e., HumanEval [7], HumanEval+ [22], MBPP [4], MBPP+ [5] and LiveCodeBench [21]. All the evaluations are conducted in a zero-shot greedy

| Model | Base | +GRPO | +PPO | +DAPO |
|-------|------|-------|------|-------|
| Llama-3.1-8B-Instruct | 49.42 | $52.28^{+2.86}$ | $52.41^{+2.99}$ | $\mathbf{53.62}^{+4.20}$ |
| OpenO1-Llama-8B-v0.1 | 52.73 | $55.63^{+2.90}$ | $54.12^{+1.39}$ | $\mathbf{60.33}^{+7.60}$ |
| Qwen2-Math-7B-Instruct | 74.46 | $74.94^{+0.46}$ | $74.93^{+0.47}$ | $\mathbf{75.41}^{+0.95}$ |
| Qwen2.5-Math-7B-Instruct | 83.42 | $84.33^{+0.91}$ | $83.76^{+0.34}$ | $\mathbf{84.86}^{+1.44}$ |

Table 3: Comparison with baseline methods on the 5K test samples from MATH [18]. DAPO consistently outperforms both the initial model and model optimized with GRPO and PPO.

sampling (i.e. temperature 0) with a cot prompt template and a maximum amount of 2048 newly generated tokens (except 4096 for OpenO1-Llama-8B-v0.1 and Skywork-O1-Open-Llama3.1-8B).

**DAPO implementations.** For each DAPO experiment, we first use the base model to generate 32 solutions for each training problem and then 6 solutions are selected out of them while making the correct solutions and wrong solutions as balanced as possible. For each selected solution, we use one line break \n to segment the solution steps and use the base model as the completer to do 16 completions for each reasoning step. Then the MC estimations are constructed for the critic training. The training methodology of critic is presented in Section 3.2. In our implementation, the critics are fine-tuned for one epoch on Qwen2.5-Math-7B-Instruct for mathematical experiments and Qwen2.5-coder-7B-Instruct for coding experiments, with a learning rate of 5e-6 and batch size 512. After that, we extract the next steps (action) for each intermediate step (state) in the completion datasets and compute the advantage using the critic for DAPO training. For all experiments, we use the global batch size 2048, learning rate 5e-7 and $\beta = 0.01$ or 0.02. We also summarize some useful tricks applied in all of the DAPO training in Appendix F.1

**Baseline methods.** We compare the performance of DAPO with PPO as the actor-critic method baseline and GRPO as the response-level method baseline on mathematical tasks using the 7.5K problems from MATH [18] as the training queries. We conduct experiments on four base models : Llama-3.1-8B-Instruct, OpenO1-Llama-8B-v0.1,Qwen2-Math-7B-Instruct and Qwen2.5-Math-7B-Instruct. We train PPO and GRPO until the training accuracy curve converges. The more training details of our baseline methods are in Appendix F.2.

## 4.2 Main results

**Math.** Our results, presented in Table 1, Table 3 and Table 4.2, demonstrate that DAPO consistently enhances the performance of the base model across all tested models on the in-domain benchmark MATH [18], through DAPO training, Skywork-Math-Llama, Llama-3.1-8B-Instruct, OpenO1-Llama-8B-v0.1, and Qwen2.5-72B-instruc achieve 50.54%, 53.62%, 60.27%, 84.55% greedy decoding accuracy respectively. For RL models, Qwen2-Math-7B-Instruct,Skywork-O1-Open-Llama3.1-8B and Qwen2.5-Math-7B-Instruct achieve 76.40%, 79.81%,and 84.86% greedy decoding accuracy respectively, indicating that DAPO can further enhance the performance even the base model are already trained by RL methods previously. Table 3 shows that DAPO has a greater improvement than PPO and GRPO on various base models in our experiments. In Figure 3, we present the accuracy curve on the MATH TEST during the training process of DAPO. As can be clearly observed, DAPO steadily enhances the accuracy throughout the training until it stabilizes.

**Code generation.** Our results for the code generation task are shown in Table 2. DAPO improves the coding performance of both Llama-3.1-8B-Instruct and OpenO1-Llama-8B-v0.1 on multiple wildly used benchmarks. For Llama-3.1-8B-Instruct, DAPO has increased by 3.0% and 2.4% respectively on HumanEval [7] and HumanEval+ [22], by 5.0% and 9.2% respectively on MBPP [4] and MBPP+ [5], and by 2.1% on LiveCodeBench [21]. For OpenO1-Llama-8B-v0.1, DAPO has increased by 2.5% and 3.6% respectively on HumanEval [7] and HumanEval+ [22], by 6.1% and 5.0% respectively on MBPP [4] and MBPP+ [5], and decreased by 2.4% on LiveCodeBench [21]. Overall, these results strongly demonstrate the effectiveness of DAPO.

**On the iterative DAPO.** As suggested by our theoretical results (Theorem 3.2), there still remains the room for performance improvement if $\pi_{\text{ref}}$ is not the optimal policy. Thus we conduct two iterative DAPO experiments on Skywork-Math-Llama and Qwen2-Math-7B-Instruct. At iteration 2, we use the model optimized by DAPO at iteration 1 as $\pi_{\text{ref}}$. Our experiment results are summarized in Table

4.2 and Table 10. We can conclude from both tables that DAPO can further improves the performance of Skywork-Math-Llama and Qwen2-Math-7B-Instruct.

| Model | Base | +DAPO iter1 | +DAPO iter2 |
|---|---|---|---|
| Skywork-Math-Llama | 41.90 | $46.88^{+4.98}$ | $\mathbf{50.54}^{+8.64}$ |
| Qwen2-Math-7B-Instruct | 74.46 | $75.41^{+0.95}$ | $\mathbf{76.40}^{+1.94}$ |

Table 4: Performance of iterative DAPO on 5K test samples from MATH [18]. Both Skywork-Math-Llama and Qwen2-Math-7B-Instruct show score improvement with additional iterations.

**Ablation studies of hyperparameters.** We conduct some preliminary ablation studies on the different components of DAPO. For the choice of KL coefficient $\beta$, we test different $\beta$ in the experiments on Skywork-Math-Llama and the results are reported in Table 4.2. It can be seen that a suitable choice of $\beta$ yields best improvement. We find $\beta = 0.01$ generally performs well on various models. Regarding the number of completions $n$ used for MC value estimation, we test $n = 8$ and $n = 16$ in

| Model $\setminus$ $\beta$ | Base | $\beta = 0.002$ | $\beta = 0.01$ | $\beta = 0.02$ | $\beta = 0.05$ | $\beta = 0.1$ |
|---|---|---|---|---|---|---|
| Skywork-Math-Llama | 41.90 | 44.52 | **46.88** | 46.70 | 45.56 | 44.50 |

Table 5: Performance of DAPO on 5K test samples from MATH [18] with different $\beta$.

the experiments on Qwen2-Math-7B-Instruct and the results are reported in Table 4.2. We find that the difference is not significant between $n = 8$ and $n = 16$ in our experiment. However, to make sure a larger performance gain as possible, we use $n = 16$ in all our experiments.

| Model $\setminus$ $n$ | Base | $n = 8$ | $n = 16$ |
|---|---|---|---|
| Qwen2-Math-7B-Instruct | 74.46 | 75.30 | **75.41** |

Table 6: Performance of DAPO on 5K test samples from MATH [18] with different number of completions for MC value estimation.

**On the computation cost of DAPO.** One major limitation that may hinder the efficient training-time scaling of DAPO is the high computation cost, particularly during the critic pre-training stage. We leave it as a future work for us to find a more efficient and scalable method for critic training. Please refer Section E for more discussions. That being said, given similar computation resources budget to the response-level baseline method, i.e. GRPO, we find that DAPO still outperforms GRPO in our experiment. We run GRPO experiment on Meta-Llama-3.1-8B-Instruct to more than 1000 training steps. The training and test curves can be found in Figure 4. It can be observed that the model exhibits serve overfitting after around 300 steps and the test performance begins to drop quickly. In contrast, as shown in Figure 3, DAPO improves the test performance steadily without exhibiting overfitting, ultimately achieving better performance gains compared to GRPO. This validates the effectiveness of granular policy optimization enabled by estimating step-level advantages.

## 5  Conclusion

In this work, we propose an offline step-level RLHF method called Direct Advantage-Based Policy Optimization (DAPO), which aims to optimize the generation of reasoning steps. DAPO significantly improves performance on both mathematical and coding benchmarks, demonstrating its effectiveness. Compared with response-level algorithms, DAPO leverages the critic function for more fine-grained policy optimization. Compared with standard actor-critic methods, DAPO separates the training of the actor and critic into two distinct stages, stabilizing the RL training process while obtaining a good value function estimation. As we discussed before, the main limitation of DAPO is the high computation cost, indicating a need to find a more efficient implementation method. Besides, it also remains a great interest to see whether using a larger set of high-quality training queries would yield greater performance improvements.

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

# A    Related work

By Theorem B.2, the the optimal policy $\pi_\beta^*$ has the analytic solution given by

$$\forall x, y : \log \frac{\pi_\beta^*(y|x)}{\pi_{\text{ref}}(y|x)} = \frac{1}{\beta}\left(r(x,y) - \hat{V}_{\text{ref}}(x)\right), \tag{16}$$

where

$$\hat{V}_{\text{ref}}(x) := \beta \log\left\{\mathbb{E}_{y' \sim \pi_{\text{ref}}(\cdot|x)}\left[\exp\left\{\frac{1}{\beta}r(x,y')\right\}\right]\right\}.$$

In order to solve $\pi_\beta^*$, DPO [42] use the Bradley-Terry model to parametrize the reward model $r$ and infers the optimal policy directly from pair-wise data without accessing to $\hat{V}_{\text{ref}}$. DRO [43] takes a straight way and solves the optimal policy directly by

$$\min_\theta \mathbb{E}_{x,y}\left[\left(\log \frac{\pi_\theta(y|x)}{\pi_{\text{ref}}(y|x)} - \frac{1}{\beta}\left(r(x,y) - \hat{V}_{\text{ref}}(x)\right)\right)^2\right].$$

one-step However, when it comes to the multi-step RLHF problem (4), the decision horizon $T$ is generally larger than one. In this case, by Theorem B.2, the optimal policy $\pi_\beta^*$ satisfies

$$\log \frac{\pi_\beta^*(a|s)}{\pi_{\text{ref}}(a|s)} = \frac{1}{\beta}\left(r(s,a) + V^*(s') - V_\beta^*(s)\right), \tag{17}$$

where $s$ is the current state, $a$ is the action, $s' = f(s,a)$ is the successive state and $V_\beta^*$ is the optimal value function of $\pi_\beta^*$ defined in (3) that can be accessed in advance. Several policy gradient ascent [49] based methods are applied to solve the optimal policy. PPO use GAE [45] to estimate the policy gradient and a clip trick to keep the training process stable. Considering the computation efficiency, GRPO [48], RLOO [2], Remax [30] remove the critic component and use the sampling rewards to estimate the gradient. [26, 37, 14] adapt the DPO algorithm to the step-level setting, developing a series of methods known as step-DPO. These methods also uses the BT model to parametrize the reward model and infer the optimal policy from (17) without the knowledge of $V_\beta^*$. However, these methods still requires the collection of massive pairwise trajectory data. DQO [32] takes the standard actor-critic route like PPO and solves the optimal policy using the Soft Actor-Critic [15](SAC) Method. WebRL [41] directly solves equation (17) by using a MSE loss and replacing $V^*$ with the value function of some known behavior policy $\mu$. OREO [54] also proposes an offline actor-critic methods for LLM multi-step reasoning. [58] and TPO [31] uses MCTS and implicit reward respectively to provide dense signals for policy optimization. [47] use the advantage function of an additional prover policy $\mu$ as a PRM in the online policy gradient. In the below, we give a holistic comparison between DAPO and other methods in Table 7.

| Method | DPO | GRPO | PPO | DQO | OREO | [58] | DAPO |
|---|---|---|---|---|---|---|---|
| Step-wise optimization | × | × | ✓ | ✓ | ✓ | ✓ | ✓ |
| Step signal provider | NA | NA | critic | critic | critic | MCTS | critic |
| Learn from listwise samples | × | ✓ | ✓ | ✓ | ✓ | × | ✓ |
| interwoven update of actor-critic | NA | NA | × | × | × | NA | ✓ |

Table 7: Comparison between DAPO and other methods

A significant difference between DAPO and other actor-critic methods is that DAPO decouples the training of the actor and critic into two distinct stages. This separation offers the following advantages during the actor training stage: **Enhanced training stability.** The non-stationary critic along with the interwoven update between actor and critic bring more instability to the training process. DAPO address this issue by removing the critic in the actor training stage. **Substantially reduced memory usage.** During the actor training stage, DAPO only loads the actor model and the advantage dataset, leading to significantly lower GPU memory consumption compared to PPO, DQO, and OREO.

# B More on MDP

**Bellman Optimality Operators**  We define the Bellman optimality operator $\mathcal{T}_\beta$ as

$$\forall s \in \mathcal{S} : [\mathcal{T}_\beta V](s) := \max_{\pi \in \Pi} \; [\mathcal{T}_\beta^\pi V](s)$$

**Lemma B.1.** *Given any function* $V : \mathcal{S} \to \mathbb{R}$,

$$\forall s \in \mathcal{S} : [\mathcal{T}_\beta V](s) = \beta \log \mathbb{E}_{a \sim \pi_{\text{ref}}(\cdot|s)} \left[ \exp \left\{ \frac{1}{\beta} \left( r(s, a) + V(f(s, a)) \right) \right\} \right] \tag{18}$$

**Optimal policy**  In the following we give the optimality conditions of the multi-step RL.

**Theorem B.2** ([1, 34]). *if a policy* $\hat{\pi} \in \Pi$ *satisfies*

$$\forall s \in \mathcal{S} : V_\beta^{\hat{\pi}}(s) = [\mathcal{T}_\beta V_\beta^{\hat{\pi}}](s),$$

*then* $\hat{\pi} = \pi_\beta^*$. *Besides, the optimal policy* $\pi_\beta^*$ *has the unique analytic form of*

$$\forall s \in \mathcal{S}, a \in \mathcal{A} : \log \frac{\pi_\beta^*(a|s)}{\pi_{\text{ref}}(a|s)} = \frac{1}{\beta} \left( Q_\beta^*(s, a) - V_\beta^*(s) \right).$$

# C Proofs

## C.1 Proof of Lemma B.1

By the definition of $\mathcal{T}_\beta$, for any state $s \in \mathcal{S}$, we define

$$\pi_\beta^V(\cdot|s) \in \operatorname*{arg\,max}_{p \in \Delta(\mathcal{A})} \mathbb{E}_{a \sim p} \left[ r(s, a) + V(f(s, a)) - \beta \log \frac{p(a)}{\pi_{\text{ref}}(a|s)} \right].$$

One can easily find that

$$\forall s \in \mathcal{S}, a \in \mathcal{A} : \pi_\beta^V(a|s) := \frac{\pi_{\text{ref}}(a|s) \cdot \exp \left\{ \frac{1}{\beta} \left( r(s, a) + V(f(s, a)) \right) \right\}}{\mathbb{E}_{a' \sim \pi_{\text{ref}}(\cdot|s)} \left[ \exp \left\{ \frac{1}{\beta} \left( r(s, a') + V(f(s, a)') \right) \right\} \right]}$$

and we omit the proof of it here. Then

$$\mathcal{T}_\beta V(s) = \mathcal{T}_\beta^{\pi_\beta^V} V(s)$$

$$= \mathbb{E}_{a \sim \pi_\beta^V(\cdot|s)} \left[ r(s, a) + V(f(s, a)) - \beta \log \frac{\pi_\beta^V(a|s)}{\pi_{\text{ref}}(a|s)} \right]$$

$$= \mathbb{E}_{a \sim \pi_\beta^V(\cdot|s)} \left[ r(s, a) + V(f(s, a)) - \beta \log \left( \frac{\exp \left\{ \frac{1}{\beta} \left( r(s, a) + V(f(s, a)) \right) \right\}}{\mathbb{E}_{a' \sim \pi_{\text{ref}}(\cdot|s)} \left[ \exp \left\{ \frac{1}{\beta} \left( r(s, a') + V(f(s, a)') \right) \right\} \right]} \right) \right]$$

$$= \beta \log \left( \mathbb{E}_{a \sim \pi_{\text{ref}}(\cdot|s)} \left[ \exp \left\{ \frac{1}{\beta} \left( r(s, a) + V(f(s, a)) \right) \right\} \right] \right)$$

## C.2 Proof of Lemma 2.2

*Proof.* Recall the definition of state value functions in (3). Then one has

$$V_\beta^\pi(\rho) - V_\beta^{\tilde{\pi}}(\rho) = \mathbb{E}_{\forall t \in \mathbb{N}: a_t \sim \pi(\cdot|s_t)} \left[ \sum_{t=0}^{T-1} \left( r(s_t, a_t) - \beta \text{KL} \left( \pi(\cdot|s_t), \pi_{\text{ref}}(\cdot|s_t) \right) \right) | s_0 \sim \rho \right] - V_\beta^{\tilde{\pi}}(\rho)$$

$$= \mathbb{E}_{\forall t \in \mathbb{N}: a_t \sim \pi(\cdot|s_t)} \left[ \sum_{t=0}^{T-1} \left( r\left(s_t, a_t\right) - \beta \mathrm{KL}\left(\pi\left(\cdot|s_t\right), \pi_{\mathrm{ref}}\left(\cdot|s_t\right)\right)\right) | s_0 \sim \rho \right]$$

$$- \mathbb{E}_{\forall t \in \mathbb{N}: a_t \sim \pi(\cdot|s_t)} \left[ \sum_{t=0}^{T-1} \left( V_\beta^{\tilde{\pi}}\left(s_t\right) - V_\beta^{\tilde{\pi}}\left(s_{t+1}\right)\right) | s_0 \sim \rho \right]$$

$$= \mathbb{E}_{\forall t \in \mathbb{N}: a_t \sim \pi(\cdot|s_t)} \left[ \sum_{t=0}^{T-1} \left( r\left(s_t, a_t\right) - \beta \mathrm{KL}\left(\pi\left(\cdot|s_t\right), \pi_{\mathrm{ref}}\left(\cdot|s_t\right)\right) + V_\beta^{\tilde{\pi}}\left(s_{t+1}\right) - V_\beta^{\tilde{\pi}}\left(s_t\right)\right) | s_0 \sim \rho \right]$$

$$= \mathbb{E}_{\forall t \in \mathbb{N}: a_t \sim \pi(\cdot|s_t)} \left[ \sum_{t=0}^{T-1} \left( \mathcal{T}_\beta^\pi V_\beta^{\tilde{\pi}}\left(s_t\right) - V_\beta^{\tilde{\pi}}\left(s_t\right)\right) | s_0 \sim \rho \right].$$

Note that under assumption 2.1, the state space $\mathcal{S}$ is finite. Thus

$$V_\beta^\pi(\rho) - V_\beta^{\tilde{\pi}}(\rho) = \mathbb{E}_{\forall t \in \mathbb{N}: a_t \sim \pi(\cdot|s_t)} \left[ \sum_{t=0}^{T-1} \left( \mathcal{T}_\beta^\pi V_\beta^{\tilde{\pi}}\left(s_t\right) - V_\beta^{\tilde{\pi}}\left(s_t\right)\right) \left( \sum_{s \in \mathcal{S}} \mathbb{I}\left\{s_t = s\right\}\right) | s_0 \sim \rho \right]$$

$$= \mathbb{E}_{\forall t \in \mathbb{N}: a_t \sim \pi(\cdot|s_t)} \left[ \sum_{s \in \mathcal{S}} \sum_{t=0}^{T-1} \left( \mathcal{T}_\beta^\pi V_\beta^{\tilde{\pi}}\left(s_t\right) - V_\beta^{\tilde{\pi}}\left(s_t\right)\right) \mathbb{I}\left\{s_t = s\right\} | s_0 \sim \rho \right]$$

$$= \mathbb{E}_{\forall t \in \mathbb{N}: a_t \sim \pi(\cdot|s_t)} \left[ \sum_{s \in \mathcal{S}} \sum_{t=0}^{T-1} \left( \mathcal{T}_\beta^\pi V_\beta^{\tilde{\pi}}\left(s\right) - V_\beta^{\tilde{\pi}}\left(s\right)\right) \mathbb{I}\left\{s_t = s\right\} | s_0 \sim \rho \right]$$

$$= \mathbb{E}_{\forall t \in \mathbb{N}: a_t \sim \pi(\cdot|s_t)} \left[ \sum_{s \in \mathcal{S}} \mathcal{T}_\beta^\pi V_\beta^{\tilde{\pi}}\left(s\right) - V_\beta^{\tilde{\pi}}\left(s\right) \sum_{t=0}^{T-1} \mathbb{I}\left\{s_t = s\right\} | s_0 \sim \rho \right]$$

$$= \sum_{s \in \mathcal{S}} \left( \mathcal{T}_\beta^\pi V_\beta^{\tilde{\pi}}\left(s\right) - V_\beta^{\tilde{\pi}}\left(s\right)\right) \cdot \mathbb{E}_{\forall t \in \mathbb{N}: a_t \sim \pi(\cdot|s_t)} \left[ \sum_{t=0}^{T-1} \mathbb{I}\left\{s_t = s\right\} | s_0 \sim \rho \right]$$

$$\overset{(a)}{=} \sum_{s \in \mathcal{S}} d_\rho^\pi(s) \cdot \left( \mathcal{T}_\beta^\pi V_\beta^{\tilde{\pi}}\left(s\right) - V_\beta^{\tilde{\pi}}\left(s\right)\right)$$

$$= \mathbb{E}_{s \sim d_\rho^\pi} \left[ \mathcal{T}_\beta^\pi V_\beta^{\tilde{\pi}}\left(s\right) - V_\beta^{\tilde{\pi}}\left(s\right)\right],$$

where (a) leverages the definition of visitation measure $d_\rho^\pi$. $\qquad \square$

### C.3 Proof of Lemma 3.1

*Proof.* Recall that

$$\mathcal{H}_s^k\left(\pi_\theta, \pi_{\mathrm{ref}}\right) = \frac{1}{2} \mathbb{E}_{a \sim \pi_{\theta_k}(\cdot|s)} \left[ \left( \frac{1}{\beta} A^{\pi_{\mathrm{ref}}}\left(s, a\right) - \log \frac{\pi_\theta\left(a|s\right)}{\pi_{\mathrm{ref}}\left(a|s\right)}\right)^2 \right]$$

A direct computation yields that

$$\nabla_\theta \mathcal{H}_s^k\left(\pi_\theta, \pi_{\mathrm{ref}}\right) = \frac{1}{2} \mathbb{E}_{a \sim \pi_{\theta_k}(\cdot|s)} \left[ \nabla_\theta \left( \frac{1}{\beta} A^{\pi_{\mathrm{ref}}}\left(s, a\right) - \log \frac{\pi_\theta\left(a|s\right)}{\pi_{\mathrm{ref}}\left(a|s\right)}\right)^2 \right]$$

$$= \mathbb{E}_{a \sim \pi_{\theta_k}(\cdot|s)} \left[ \left( \frac{1}{\beta} A^{\pi_{\mathrm{ref}}}\left(s, a\right) - \log \frac{\pi_\theta\left(a|s\right)}{\pi_{\mathrm{ref}}\left(a|s\right)}\right) \nabla_\theta \left( \frac{1}{\beta} A^{\pi_{\mathrm{ref}}}\left(s, a\right) - \log \frac{\pi_\theta\left(a|s\right)}{\pi_{\mathrm{ref}}\left(a|s\right)}\right) \right]$$

$$= -\mathbb{E}_{a \sim \pi_{\theta_k}(\cdot|s)} \left[ \left( \frac{1}{\beta} A^{\pi_{\mathrm{ref}}}\left(s, a\right) - \log \frac{\pi_\theta\left(a|s\right)}{\pi_{\mathrm{ref}}\left(a|s\right)}\right) \nabla_\theta \log \pi_\theta\left(a|s\right) \right].$$

Note that

$$\mathcal{I}_s\left(\pi_\theta, \pi_{\mathrm{ref}}\right) = \mathbb{E}_{a \sim \pi_\theta(\cdot|s)} \left[ A^{\pi_{\mathrm{ref}}}\left(s, a\right) - \beta \log \frac{\pi_\theta\left(a|s\right)}{\pi_{\mathrm{ref}}\left(a|s\right)}\right].$$

Then

$$\nabla_\theta \mathcal{I}_s \left( \pi_\theta, \pi_{\mathrm{ref}} \right) = \nabla_\theta \mathbb{E}_{a \sim \pi_\theta(\cdot|s)} \left[ A^{\pi_{\mathrm{ref}}} \left( s, a \right) - \beta \log \frac{\pi_\theta \left( a|s \right)}{\pi_{\mathrm{ref}} \left( a|s \right)} \right]$$

$$= \mathbb{E}_{a \sim \pi_\theta(\cdot|s)} \left[ \nabla_\theta \log \pi_\theta \left( a|s \right) \left( A^{\pi_{\mathrm{ref}}} \left( s, a \right) - \beta \log \frac{\pi_\theta \left( a|s \right)}{\pi_{\mathrm{ref}} \left( a|s \right)} \right) \right]$$

$$+ \mathbb{E}_{a \sim \pi_\theta(\cdot|s)} \left[ \nabla_\theta \left( A^{\pi_{\mathrm{ref}}} \left( s, a \right) - \beta \log \frac{\pi_\theta \left( a|s \right)}{\pi_{\mathrm{ref}} \left( a|s \right)} \right) \right]$$

$$= \beta \mathbb{E}_{a \sim \pi_\theta(\cdot|s)} \left[ \nabla_\theta \log \pi_\theta \left( a|s \right) \left( \frac{1}{\beta} A^{\pi_{\mathrm{ref}}} \left( s, a \right) - \log \frac{\pi_\theta \left( a|s \right)}{\pi_{\mathrm{ref}} \left( a|s \right)} \right) \right]$$

$$- \beta \mathbb{E}_{a \sim \pi_\theta(\cdot|s)} \left[ \nabla_\theta \log \pi_\theta \left( a|s \right) \right]$$

$$\overset{(a)}{=} \beta \mathbb{E}_{a \sim \pi_\theta(\cdot|s)} \left[ \nabla_\theta \log \pi_\theta \left( a|s \right) \left( \frac{1}{\beta} A^{\pi_{\mathrm{ref}}} \left( s, a \right) - \log \frac{\pi_\theta \left( a|s \right)}{\pi_{\mathrm{ref}} \left( a|s \right)} \right) \right],$$

where (a) is due to

$$\mathbb{E}_{a \sim \pi_\theta(\cdot|s)} \left[ \nabla_\theta \log \pi_\theta \left( a|s \right) \right] = \nabla_\theta \mathbb{E}_{a \sim \pi_\theta(\cdot|s)} \left[ 1 \right] = \nabla_\theta \left( 1 \right) = 0.$$

Thus it's easy to verify that $\nabla_\theta \mathcal{I}_s \left( \pi_{\theta_k}, \pi_{\mathrm{ref}} \right) = -\beta \nabla_\theta \mathcal{H}_s^k \left( \pi_\theta, \pi_{\mathrm{ref}} \right)$ and the proof is completed.

$\square$

### C.4  Proof of Theorem 3.2

*Proof.* Notice that $\forall s \in \mathcal{S}, \nu_{\mathcal{S}}(s) > 0$. Then it's obviously that for any state $s \in \mathcal{S}$,

$$\pi^+ \left( \cdot | s \right) \in \underset{p \in \Delta(\mathcal{A})}{\arg \min} \, \mathbb{E}_{a \sim \nu_{\mathcal{A}}(\cdot|s)} \left[ \left( \frac{1}{\beta} A^{\pi_{\mathrm{ref}}} \left( s, a \right) - \log \frac{p \left( a \right)}{\pi_{\mathrm{ref}} \left( a|s \right)} \right)^2 \right].$$

Since $\forall s \in \mathcal{S}, a \in \mathcal{A}, \nu_{\mathcal{A}}(a|s) > 0, \pi_{\mathrm{ref}}(a|s) > 0$, one has $\forall s \in \mathcal{S}, a \in \mathcal{A}, \pi^+(a|s) > 0$ otherwise the loss becomes infinity. For ease of notation, we define

$$u_{s,a} := \log \frac{p \left( a \right)}{\pi_{\mathrm{ref}} \left( a|s \right)} \qquad \text{and} \qquad A_{s,a} := \frac{1}{\beta} A^{\pi_{\mathrm{ref}}} \left( s, a \right).$$

Then $\left\{ u_{s,a}^+ := \log \frac{\pi^+(a|s)}{\pi_{\mathrm{ref}}(a|s)} \mid a \in \mathcal{A} \right\}$ is the solution of the following optimization problem:

$$\underset{u_s := (u_{s,a})_{a \in \mathcal{A}} \in \mathbb{R}^{|\mathcal{A}|}}{\min} : \quad \mathcal{L}_s \left( u_s \right) := \frac{1}{2} \cdot \mathbb{E}_{a \sim \nu_{\mathcal{A}}(\cdot|s)} \left[ \left( A_{s,a} - u_{s,a} \right)^2 \right]$$

$$\text{s.t} : \quad \sum_{a \in \mathcal{A}} p \left( a \right) = \mathbb{E}_{a \sim \pi_{\mathrm{ref}}(\cdot|s)} \left[ \exp \left\{ u_{s,a} \right\} \right] = 1.$$

Denote the Lagrangian function as

$$\hat{\mathcal{L}}_s \left( u_s, \lambda \right) := \mathcal{L}_s \left( u_s \right) + \lambda \cdot \left( \mathbb{E}_{a \sim \pi_{\mathrm{ref}}(\cdot|s)} \left[ \exp \left\{ u_{s,a} \right\} \right] - 1 \right).$$

Since it's a convex problem, by KKT conditions, there exists a $\lambda_s^* \in \mathbb{R}$ such that $(u_s^+, \lambda_s^*)$ is the solution of the equations:

$$\begin{cases} \nabla_{u_s} \hat{\mathcal{L}}_s \left( u_s^+, \lambda_s^* \right) = 0 \\ \mathbb{E}_{a \sim \pi_{\mathrm{ref}}(\cdot|s)} \left[ \exp \left\{ u_{s,a}^+ \right\} \right] = 1 \end{cases} \tag{19}$$

Notice that

$$\nabla_{u_s} \hat{\mathcal{L}}_s \left( u_s, \lambda \right) = \nabla_{u_s} \mathbb{E}_{a \sim \nu_{\mathcal{A}}(\cdot|s)} \left[ \frac{1}{2} \left( A_{s,a} - u_{s,a} \right)^2 + \lambda \cdot \frac{\pi_{\mathrm{ref}} \left( a|s \right)}{\nu \left( a|s \right)} \cdot \exp \left\{ u_{s,a} \right\} \right]$$

$$= \nu_{\mathcal{A}} \left( \cdot | s \right) \odot \left( u_s - A_s + \lambda \cdot \frac{\pi_{\mathrm{ref}} \left( \cdot | s \right)}{\nu_{\mathcal{A}} \left( \cdot | s \right)} \cdot \exp \left\{ u_s \right\} \right).$$

Plugging it into (19) yields that

$$\forall a \in \mathcal{A} : A_{s,a} - u_{s,a}^+ = \lambda_s^* \cdot \frac{\pi_{\mathrm{ref}}\left(a|s\right)}{\nu_{\mathcal{A}}\left(a|s\right)} \cdot \exp\left\{u_{s,a}^+\right\}. \tag{20}$$

A direct computation yields that

$$\begin{aligned}
\lambda_s^* &= \lambda_s^* \cdot \mathbb{E}_{a \sim \pi_{\mathrm{ref}}}\left[\exp\left\{u_{s,a}^+\right\}\right] \\
&\overset{(a)}{=} \mathbb{E}_{a \sim \nu_{\mathcal{A}}(\cdot|s)}\left[\lambda_s^* \cdot \frac{\pi_{\mathrm{ref}}\left(a|s\right)}{\nu\left(a|s\right)} \cdot \exp\left\{u_{s,a}^+\right\}\right] \\
&= \mathbb{E}_{a \sim \nu_{\mathcal{A}}(\cdot|s)}\left[A_{s,a} - u_{s,a}^+\right],
\end{aligned}$$

where (a) is due to (20). We now to show that $\lambda_s^* \geq 0$. Notice that

$$\begin{aligned}
\mathbb{E}_{a \sim \pi_{\mathrm{ref}}(\cdot|s)}\left[A_{s,a} - u_{s,a}^+\right] &= \mathbb{E}_{a \sim \pi_{\mathrm{ref}}(\cdot|s)}\left[\frac{1}{\beta}A^{\pi_{\mathrm{ref}}}\left(a|s\right) - \log\frac{\pi^+\left(a|s\right)}{\pi_{\mathrm{ref}}\left(a|s\right)}\right] \\
&= \mathbb{E}_{a \sim \pi_{\mathrm{ref}}(\cdot|s)}\left[\log\frac{\pi_{\mathrm{ref}}\left(a|s\right)}{\pi^+\left(a|s\right)}\right] \\
&= \mathrm{KL}\left(\pi_{\mathrm{ref}}\left(\cdot|s\right), \pi^+\left(\cdot|s\right)\right). \tag{21}
\end{aligned}$$

Thus if $\lambda_s^* < 0$, one has $\forall a \in \mathcal{A} : A_{s,a} - u_{s,a}^+ < 0$ due to (20) and

$$\mathbb{E}_{a \sim \pi_{\mathrm{ref}}}\left[A_{s,a} - u_{s,a}^+\right] < 0,$$

which contradicts to (21). Recall the definition of $\mathcal{I}_s(\pi, \pi_{\mathrm{ref}})$. Thus

$$\begin{aligned}
\mathcal{I}_s\left(\pi^+, \pi_{\mathrm{ref}}\right) &= \mathbb{E}_{a \sim \pi^+(\cdot|s)}\left[A^{\pi_{\mathrm{ref}}}\left(s,a\right) - \beta\log\frac{\pi^+\left(a|s\right)}{\pi_{\mathrm{ref}}\left(a|s\right)}\right] \\
&= \beta \cdot \mathbb{E}_{a \sim \pi^+(\cdot|s)}\left[A_{s,a} - u_{s,a}^+\right] \\
&\overset{(a)}{=} \beta \cdot \mathbb{E}_{a \sim \pi^+(\cdot|s)}\left[\lambda_s^* \cdot \frac{\pi_{\mathrm{ref}}\left(a|s\right)}{\nu_{\mathcal{A}}\left(a|s\right)} \cdot \exp\left\{u_{s,a}^+\right\}\right] \\
&= \beta \cdot \lambda_s^* \cdot \mathbb{E}_{a \sim \pi^+(\cdot|s)}\left[\frac{\pi^+\left(a|s\right)}{\nu_{\mathcal{A}}\left(a|s\right)}\right] \\
&= \beta \cdot \lambda_s^* \cdot \mathbb{E}_{a \sim \nu_{\mathcal{A}}(\cdot|s)}\left[\left(\frac{\pi^+\left(a|s\right)}{\nu_{\mathcal{A}}\left(a|s\right)}\right)^2\right] \\
&= \beta \cdot \lambda_s^* \cdot \mathbb{E}_{a \sim \nu_{\mathcal{A}}(\cdot|s)}\left[\left(\frac{\pi^+\left(a|s\right)}{\nu_{\mathcal{A}}\left(a|s\right)}\right)^2\right] \\
&\geq \beta \cdot \lambda_s^* \\
&:= \lambda_s(\nu_{\mathcal{A}}) \tag{22}
\end{aligned}$$

where the last inequality is due to

$$\mathbb{E}_{a \sim \nu_{\mathcal{A}}(\cdot|s)}\left[\left(\frac{\pi^+\left(a|s\right)}{\nu_{\mathcal{A}}\left(a|s\right)}\right)^2\right] \geq \left(\mathbb{E}_{a \sim \nu_{\mathcal{A}}(\cdot|s)}\left[\frac{\pi^+\left(a|s\right)}{\nu_{\mathcal{A}}\left(a|s\right)}\right]\right)^2 = \left(\sum_{a \in \mathcal{A}}\pi^+\left(a|s\right)\right)^2 = 1.$$

Plugging (22) into Lemma 2.2 directly yields that

$$V_\beta^{\pi^+}\left(\mu\right) - V_\beta^{\pi_{\mathrm{ref}}}\left(\mu\right) = \mathbb{E}_{d_\mu^{\pi^+}}\left[\mathcal{I}_s\left(\pi, \pi_{\mathrm{ref}}\right)\right] \geq \mathbb{E}_{d_\mu^{\pi^+}}\left[\lambda_s\left(\nu_{\mathcal{A}}\right)\right] \geq 0.$$

We now to show that $V_\beta^{\pi^+}\left(\mu\right) = V_\beta^{\pi_{\mathrm{ref}}}\left(\mu\right)$ holds if and only if $\pi_{\mathrm{ref}}$ is already the optimal policy. Notice that

$$V_\beta^{\pi^+}\left(\mu\right) = V_\beta^{\pi_{\mathrm{ref}}}\left(\mu\right) \;\Leftrightarrow\; \forall s \in \mathcal{S} : \lambda_s\left(\nu_{\mathcal{A}}\right) = 0 \Leftrightarrow \forall s \in \mathcal{S}, a \in \mathcal{A} : u_{s,a}^+ - A_{s,a} = 0.$$

Thus plugging it into (21) yields that

$$\mathrm{KL}\left(\pi_{\mathrm{ref}}\left(\cdot|s\right), \pi^+\left(\cdot|s\right)\right) = 0$$

and $\forall s, a \in \mathcal{S} \times \mathcal{A} : A^{\pi_{\mathrm{ref}}}(s,a) = \beta \cdot \log \frac{\pi^+(a|s)}{\pi_{\mathrm{ref}}(a|s)} = 0$. Hence for any state $s \in \mathcal{S}$,

$$
\begin{aligned}
\mathcal{T}_\beta V_\beta^{\pi_{\mathrm{ref}}}(s) &= \beta \log \mathbb{E}_{a \sim \pi_{\mathrm{ref}}(\cdot|s)} \left[ \exp\left\{ \frac{1}{\beta} \left( r(s,a) + V_\beta^{\pi_{\mathrm{ref}}}(f(s,a)) \right) \right\} \right] \\
&= \beta \log \mathbb{E}_{a \sim \pi_{\mathrm{ref}}(\cdot|s)} \left[ \exp\left\{ \frac{1}{\beta} Q_\beta^{\pi_{\mathrm{ref}}}(s,a) \right\} \right] \\
&= \beta \log \mathbb{E}_{a \sim \pi_{\mathrm{ref}}(\cdot|s)} \left[ \exp\left\{ \frac{1}{\beta} Q^{\pi_{\mathrm{ref}}}(s,a) \right\} \right] \\
&= \beta \log \mathbb{E}_{a \sim \pi_{\mathrm{ref}}(\cdot|s)} \left[ \exp\left\{ \frac{1}{\beta} A^{\pi_{\mathrm{ref}}}(s,a) + \frac{1}{\beta} V^{\pi_{\mathrm{ref}}}(s) \right\} \right] \\
&= \beta \log \mathbb{E}_{a \sim \pi_{\mathrm{ref}}(\cdot|s)} \left[ \exp\left\{ \frac{1}{\beta} V^{\pi_{\mathrm{ref}}}(s) \right\} \right] \\
&= V^{\pi_{\mathrm{ref}}}(s) \\
&= V_\beta^{\pi_{\mathrm{ref}}}(s)
\end{aligned}
$$

which means $V_\beta^{\pi_{\mathrm{ref}}}$ is the fixed point of the Bellman optimality operator thus $\pi_{\mathrm{ref}}$ is the optimal policy by Theorem B.2 □

## D  Discussion I : connection between DAPO and multi-step reasoning.

In this section, we give a detailed discussion on the connections between DAPO and multi-step reasoning and show that DAPO indeed achieves step-level fine-grained policy optimization and reduces the training variance.

**Response-level policy gradient methods suffer from high training variance.** Policy gradient (PG) methods updates the parameter by applying gradient ascent w.r.t the objective (4) directly and generally takes the form of

$$
\theta \leftarrow \theta + \eta \nabla_\theta V_\beta^\pi(\mu).
$$

For simplicity of analysis, we assume $\beta = 0$. By policy gradient theorem [1],

$$
\nabla_\theta V^{\pi_\theta}(\mu) = \mathbb{E}_{x \sim \mathcal{D}} \mathbb{E}_{\forall t : a_t \sim \pi_\theta(\cdot|x)} \left[ \sum_{t=0}^{T-1} \nabla_\theta \log \pi_\theta(a_t|s_t) A^{\pi_\theta}(s_t, a_t) \right]. \tag{23}
$$

Current response-level PG methods mainly focus on the estimation of advantage in (23). Suppose $n$ responses $y_1, ..., y_n$ are sampled from $\pi_\theta(\cdot|x)$ for each prompt $x$. We denote $y_i := \mathrm{Concat}(x, a_0^{(i)}, ..., a_{T_i-1}^{(i)})$, $s_t^{(i)} := \mathrm{Concat}(x, a_0^{(i)}, ..., a_{t-1}^{(i)})$, where $T_i$ is the number of reasoning steps in response $y_i$, and $A_t^{(i)}$ as the estimation of advantage $A^{\pi_\theta}(s_t^{(i)}, a_t^{(i)})$. We summarize the construction of $A_t^{(i)}$ of different methods in Table 8. It can be observed that all methods use $r(x, y_i)$

| Methods | Advantage estimation $A_t^{(i)}$ | Sign of $A_t^{(i)}$ | Does $A_t^{(i)}$ vary with $t$ ? |
|---|---|---|---|
| REINFORCE | $r(x,y)$ | non-negative when $r(x,y_i)=1$ and non-positive when $r(x,y_i)=0$ | No |
| REINFORCE++ [19] | $r(x,y)$ with global batch normalization | non-negative when $r(x,y_i)=1$ and non-positive when $r(x,y_i)=0$ | No |
| ReMax [30] | $r(x,y_i) - r(x,\hat{y})$ where $\hat{y}_i$ is sampled from the greedy policy | non-negative when $r(x,y_i)=1$ and non-positive when $r(x,y_i)=0$ | No |
| GRPO [48] | $\frac{r(x,y_i)-\hat{p}}{\sqrt{\hat{p}(1-\hat{p})}}$, where $\hat{p} := \frac{1}{n}\sum_{j=1}^n r(x,y_j)$ | non-negative when $r(x,y_i)=1$ and non-positive when $r(x,y_i)=0$ | No |
| ORZ [20] | $r(x,y_i) - V_\phi\left(s_t^{(i)}\right)$ | non-negative when $r(x,y_i)=1$ and non-positive when $r(x,y_i)=0$ | No |
| DR.GRPO [36] | $r(x,y_i) - \hat{p}$, where $\hat{p} := \frac{1}{n}\sum_{j=1}^n r(x,y_j)$ | non-negative when $r(x,y_i)=1$ and non-positive when $r(x,y_i)=0$ | No |

Table 8: Summary of advantage estimation of current response-level PG methods.

as the MC estimation of state-value function . Thus these methods generally try to increase the probability of **all** the reasoning steps if the response $y$ is correct and decrease the probability of **all** the reasoning steps if $y$ is incorrect. However, this update pattern indeed introduces high training variance since the sign of the true advantage value, i.e. $A^{\pi_\theta}(s_t, a_t)$, may mismatch the final reward of the trajectory. Following is a concrete example. The trajectory shown in Figure 2 contains 14 reasoning steps and is a correct trajectory sampled from Llama-3.1-8B-instruct. The values and advantages of each reasoning step are also reported on the right-hand side of the figure. There is a noticeable drop in value (accuracy) in step 11, decreasing from 0.94 to 0.41, resulting in an advantage of -0.53. This means the model is more likely to make a mistake starting from step 11. A well-directed policy update may reduce the likelihood of step 11, rather than increasing it, even if the final answer is correct. **Thus, there exists responses that yield correct answers while containing reasoning steps from which the model is likely to make mistakes, and vice versa.** Thus, using the same advantage estimation for all reasoning steps in the entire response introduces high training variance.

| Question | Reasoning Steps | Value | Advantage |
|---|---|---|---|
| What is the volume, in cubic inches, of a rectangular box, whose faces have areas of 24 square inches, 16 square inches and 6 square inches? | Step 1 : To find the volume of a rectangular box, we need to multiply its length, width, and height. | 0.41 | 0 |
| | Step 2 : Let's call the length, width, and height of the box $l$, $w$, and $h$ respectively. We are given the areas of the faces as follows: | 0.53 | 0.12 |
| | Step 3 : - The face with area 24 has dimensions $l * w$. | 0.53 | 0 |
| | Step 4 : - The face with area 16 has dimensions $w * h$. | 0.81 | 0.28 |
| | Step 5 : - The face with area 6 has dimensions $l * h$. | 0.56 | -0.25 |
| | Step 6 : So, we have the following equations from the given areas:. | 0.53 | -0.03 |
| | Step 7 : - $lw = 24, wh = 16, lh = 6$. | 0.84 | 0.31 |
| | Step 8 : To find the volume, we can multiply all three equations together to get: | 0.84 | 0 |
| | Step 9 : $(lw)(wh)(lh) = 24 * 16 * 6$. | 0.87 | 0.03 |
| | Step 10 : This simplifies to $(lwh)^2 = 2304$. | 0.94 | 0.07 |
| | Step 11 : Taking the square root of both sides gives: $lwh = \sqrt{2304}$. | 0.41 | -0.53 |
| | Step 12 : $lwh = \sqrt{144 * 16}$. | 1 | 0.59 |
| | Step 13 : $lwh = 12 * 4 = 48$. | 1 | 0 |
| | Step 14 : The volume is $\boxed{48}$ cubic inches. | 1 | 0 |

Figure 2: A reasoning trajectory sampled from Llama-3.1-8B-instruct with the true state value and advantage reported in the right hand side.

**Step-level fine-grained policy optimization via DAPO.** Even though the objective (1) optimized by response-level algorithms can be reformulated to a multi-step decision problem (2) equivalently, it by no means implies that these methods can achieve step-level optimization as illustrated above. For DAPO, it can be directly observed from (12) that, given the previous reasoning steps (state $s$), DAPO optimizes the distribution of the next reasoning step generation (action $a$) depends on the step-level advantage estimation other than the final reward. Thus DAPO indeed enables fine-grained step-wise learning by optimizing the actor using the advantage datasets. Our main theoretical result (Theorem 3.2) further shows that updating the actor's probability in the direction of the advantage can indeed yield policy improvement if the advantages are well-estimated.

# E  Discussion II : Computational cost of DAPO

In this section, we provide a detailed analysis of the computational cost of DAPO and discuss potential approaches for efficient training-time scaling. Consider an actor model $M_{\text{actor}}$ with $N_{\text{model}}^{\text{actor}}$

non-embedding parameters is optimized by DAPO on an offline advantage dataset $\mathcal{D}$ with batch size $B_{\text{actor}}$ over $K_{\text{actor}}$ training steps. For the critic pre-training, we first generate $n$ initial responses from which $n_1$ responses are selected. For each reasoning step in these filtered responses, we then produce $m$ completions for MC value estimation. Then a critic model $M_{\text{critic}}$ with $N_{\text{model}}^{\text{critic}}$ non-embedding parameters is optimized on the MC value dataset with batch size $B_{\text{critic}}$ over $K_{\text{critic}}$ training steps. To simplify the analysis, we assume both the response length and the number of reasoning steps per response remain constant at $L$ and $T$ respectively (each reasoning step is of length $L/T$). Following [24], we estimate inference computational cost as $2N_{\text{model}}N_{\text{infer}}$ and the training computational cost can be estimated by $6N_{\text{model}}N_{\text{train}}$, where $N_{\text{infer}}$ and $N_{\text{train}}$ are inference and training tokens respectively. We report $N_{\text{infer}}$ and $N_{\text{train}}$ for our DAPO implementation in Table E.

| **Step** | $N_{\textbf{infer}}$ | $N_{\textbf{train}}$ | Computational cost $N_{\text{model}}(2N_{\text{infer}} + 6N_{\text{train}})$ |
|---|---|---|---|
| Generation | $n_1 L$ | $0$ | $2n_1 L \cdot N_{\text{model}}^{\text{actor}}$ |
| Completion | $\frac{nmL}{2}(T-1)$ | $0$ | $nmL(T-1) \cdot N_{\text{model}}^{\text{actor}}$ |
| Critic Optimization | $0$ | $nT \cdot B_{\text{critic}}K_{\text{critic}}$ | $6nT \cdot B_{\text{critic}}K_{\text{critic}} \cdot N_{\text{model}}^{\text{critic}}$ |
| Making Advantage Dataset | $nmL$ | $0$ | $2nmL \cdot N_{\text{model}}^{\text{critic}}$ |
| Actor Optimization | $0$ | $L/T \cdot B_{\text{actor}}K_{\text{actor}}$ | $6L/T \cdot B_{\text{actor}}K_{\text{actor}} \cdot N_{\text{model}}^{\text{actor}}$ |

Table 9: Computational cost estimation of DAPO.

Suppose $N_{\text{model}}^{\text{actor}} = N_{\text{model}}^{\text{critic}} = N_{\text{model}}$. By Table E, the total computational cost $c$ of DAPO is

$$c = N_{\text{model}}\left(L\left(2n_1 + nm\left(T-1\right) + 2nm\right) + 6\left(nTB_{\text{critic}}K_{\text{critic}} + \frac{L}{T}B_{\text{actor}}K_{\text{actor}}\right)\right)$$

$$= N_{\text{model}} \cdot \mathcal{O}\left(LTnm + \left(\frac{L}{T} + nT\right)BK\right),$$

where $B := \max\{B_{\text{actor}}, B_{\text{critic}}\}$ and $K := \max\{K_{\text{actor}}, K_{\text{critic}}\}$. In our implementation $n_1 = 32, n = 6, m = 16, T \approx 20, L \approx 2048$.

**Possible approaches to efficient training-time scaling.** Suppose the length of reasoning step $L_{\text{step}} := \frac{L}{T}$ is a constant. Then the computation cost $c = N_{\text{model}} \cdot L_{\text{step}} \cdot \mathcal{O}\left(T^2 nm + nTBK\right)$. The computational cost $c$ may explode due to square term $T^2$ as the response length $L$ increased. A potential approach to mitigate this issue is **not** to do MC value estimation for **all** reasoning steps as in [38]. Another possible approach is to find an efficient and effective method for dividing reasoning steps, ensuring that the total number of reasoning steps $T$ does not increase too rapidly as in [35].

# F    More details of experiments

## F.1    DAPO

| **Model** | In Domain MATH | | Out of Domain | | |
|---|---|---|---|---|---|
| | | GSM8K | Minerva MATH | Olympiad Bench | College Math |
| Skywork-Math-LLama | 41.90 | 61.49 | 5.51 | 18.68 | 24.85 |
| + DAPO iter 1 | 46.88$^{+4.98}$ | 67.55$^{+6.06}$ | 7.34$^{+1.82}$ | 22.38 $^{+3.70}$ | 25.87 $^{+1.03}$ |
| + DAPO iter 2 | 50.54$^{+8.64}$ | 69.04$^{+7.55}$ | 8.09$^{+2.58}$ | 23.86 $^{+5.18}$ | 27.20 $^{+2.35}$ |
| Qwen2-Math-7B-Instruct | 74.46 | 89.38 | 40.07 | 34.36 | 41.59 |
| + DAPO iter 1 | 75.41 $^{+0.95}$ | 89.45$^{+0.07}$ | 37.51$^{-2.56}$ | 37.03 $^{+2.67}$ | 42.23 $^{+0.64}$ |
| + DAPO iter 2 | 76.40 $^{+1.94}$ | 89.30$^{-0.08}$ | 38.25$^{-1.82}$ | 37.64 $^{+3.27}$ | 41.16 $^{-0.43}$ |

Table 10: **Full Mathematical benchmark results of iterative DAPO** using zero-shot greedy inference.

**Useful training tricks**  Following are some useful tricks applied in all of the DAPO experiments.

- **State-wise learning.** Notice that for each training state (intermediate step) $s$, there are multiple actions (next step) for DAPO training. Rather than shuffling all state-action pairs, we try to place different actions from the same state within the same batch for DAPO training in order to provide contrastive gradients.

- **Learning uniformly on approximate action space.** Suppose we have $n$ actions $\{a_i\}_{i=1}^n$ for each training state $s$. Although the true action space $\mathcal{A}$ is extremely large for LLM, we consider the unique actions of $\{a_i\}_{i=1}^n$ as the approximate action space denoted as $\tilde{A}_s$. After computing the advantage for each action, we only add the unique actions to the advantage dataset for DAPO training, which means the training action distribution is $\nu_{\mathcal{A}}(\cdot|s) = \mathrm{Unif}(\tilde{A}_s)$ in our implementation.

- **Learning on the states with large advantage gap.** Considering that the approximation error between the prediction of critic network and the true state value, we only learn those states that have a large advantage gap to make sure the actions with higher advantage are actually better than those actions with a lower advantage. We define the advantage gap as

$$\Delta_s := \max_{a_i \in \tilde{A}_s} A(s, a_i) - \min_{a_j \in \tilde{A}_s} A(s, a_j).$$

In our implementation, we only learn those states with $\Delta_s >= 0.1$

## F.2    Baseline methods

| Hyperparameter | Value |
|---|---|
| actor learning rate | 5e-7 |
| critic learning rate | 9e-6 |
| KL coefficient $\beta$ | 0.01 |
| discount factor $\gamma$ | 1 |
| GAE factor $\lambda$ | 0.95 |
| actor training epoch | 1 |
| critic training epoch | 1 |
| rollout max new tokens | 2048 (4096 for OpenO1-Llama-8B-v0.1) |
| rollout temperature | 1 |
| rollout batch size | 1024 |
| global train batch size | 1024 |

Table 11: The hyperparameters we employ to train PPO

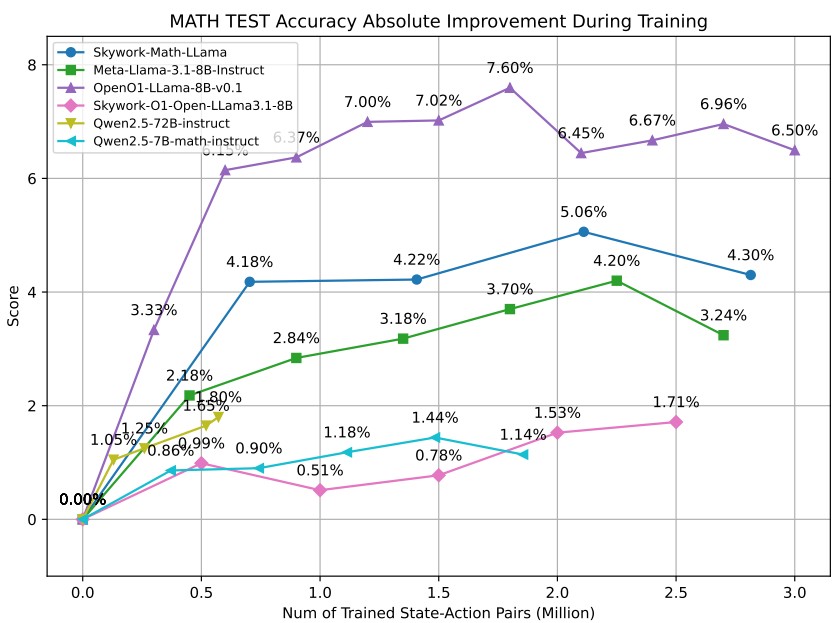

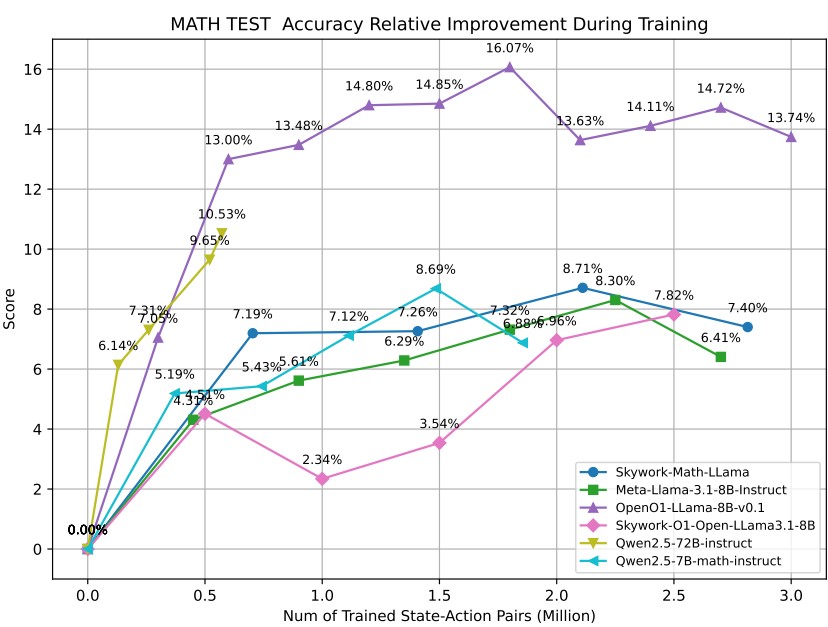

Figure 3: **Accuracy curve on MATH TEST during training process.** Let $x$ be the accuracy of base model on MATH TEST and $y$ be the accuracy after DAPO training. Absolute improvement refers to $y - x$ and relative improvement refers to $\frac{y-x}{1-x}$.

| Hyperparameter | Value |
|:---:|:---:|
| query batch size | 64 |
| group size | 16 |
| KL coefficient $\beta$ | 0.01 |
| rollout max new tokens | 2048 (4096 for OpenO1-Llama-8B-v0.1) |
| rollout temperature | 1 |
| global train batch size | 1024 |
| actor learning rate | 1e-6 |

Table 12: The hyperparameters we employ to train GRPO

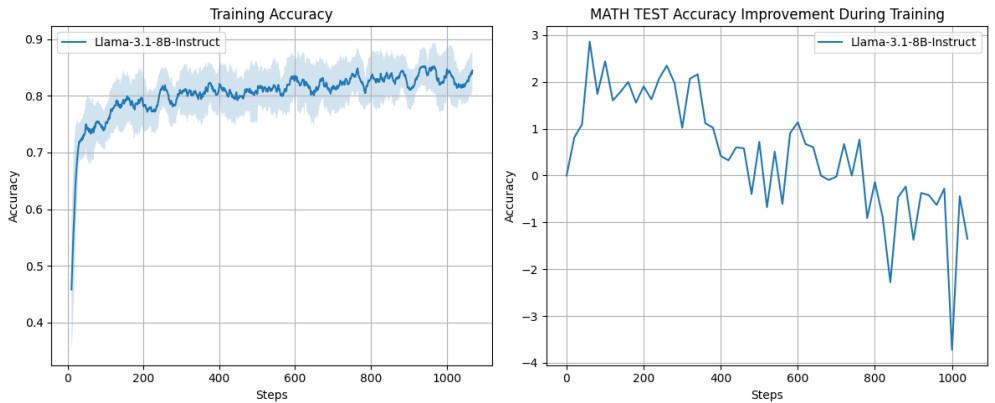

Figure 4: Performance of Meta-Llama-3.1-8B-instruct optimized via GRPO. The model exhibits serve overfitting after around 300 steps. **Left:** Accuracy on training batch during RL process. **Right:** Performance improvement (in percentage points) on MATH 5K test problems.

# G    Pseudocode of DAPO

**Algorithm 1** Direct Advantage-Based Policy Optimization

1: **Input:** Training problems $\mathcal{D}$, initial actor $\pi_{\theta_0}$, initial critic $V_{\phi_0}$, maximum iteration number $T$, KL coefficient $\beta$.

2: **for** iteration $i = 0, 1, ..., T-1$ **do**

3:    **[Step A.1 Generation]**

4:    $\mathcal{D}_{\text{gen}} \leftarrow \emptyset$, $\pi_{\text{gen}} \leftarrow \pi_{\theta_i}$, $\pi_{\text{ref}} \leftarrow \pi_{\theta_i}$.

5:    **for** problem $x \in \mathcal{D}$ **do**

6:       Sample multiple responses from the generator, i.e. $G(x) \leftarrow \{y\} \overset{\text{iid}}{\sim} \pi_{\text{gen}}(\cdot|x)$.

7:       For each response $y \in G(x)$, split it into multiple reasoning steps (e.g. by using the line break $\backslash n$) so that $y = (a_0, ..., a_{T-1})$, and the construct the states

$$\mathcal{S}(y) \leftarrow \{\text{Concat}(x, a_0, ..., a_t)\}_{t=0}^{T-1}.$$

8:       Add the states derived from all responses into $\mathcal{D}_{\text{gen}}$, i.e., $\mathcal{D}_{\text{gen}} \leftarrow \mathcal{D}_{\text{gen}} + \bigcup_{y \in G(x)} \mathcal{S}(y)$.

9:       Stop generation when certain criteria are met (e.g., when the computation budget is reached).

10:    **end for**

11:    **[Step A.2 Completion]**

12:    $\mathcal{D}_{\text{com}} \leftarrow \emptyset$, $\mathcal{D}_{\text{MC}} \leftarrow \emptyset$.

13:    **for** state $s \in \mathcal{D}_{\text{gen}}$ **do**

14:       Sample multiple completions $C(s) \leftarrow \{\tilde{y}\} \overset{\text{iid}}{\sim} \pi_{\text{ref}}(\cdot|s)$ using $s$ as the context prefix.

15:       Reward each completion and compute the MC value as

$$\text{MC}(s) \leftarrow \frac{1}{|C(s)|} \sum_{\tilde{y} \in C(s)} r(s, \tilde{y}).$$

16:       $\mathcal{D}_{\text{MC}}.\text{add}((s, \text{MC}(s)))$, $\mathcal{D}_{\text{com}}.\text{add}((s, C(s)))$.

17:    **end for**

18:    Update the critic (starting from $\phi_i$) by optimizing (15) on MC value dataset $\mathcal{D}_{\text{MC}}$, i.e.

$$\phi_{i+1} \leftarrow \arg\min_{\phi} \mathbb{E}_{s \sim \mathcal{D}_{\text{MC}}} \left[ \mathcal{L}_{\text{BCE}}(\text{MC}(s), V_\phi(s)) \right].$$

19:    **[Step B.1 Compute Advantages]**

20:    $\mathcal{D}_{\text{adv}} \leftarrow \emptyset$

21:    **for** $(s, C(s)) \in \mathcal{D}_{\text{com}}$ **do**

22:       Set action set $\mathcal{A}(s) \leftarrow \emptyset$.

23:       For each completion $\tilde{y} \in C(s)$, extract the first reasoning step $a$ from $\tilde{y}$ and add it into $A(s)$.

24:       Compute advantage for each action $a$ in $\mathcal{A}(s)$ using optimized critic, i.e.,

$$\forall a \in \mathcal{A}(s): \hat{A}(s, a) \leftarrow V_{\phi_{i+1}}(s \circ a) - \frac{1}{|\mathcal{A}(s)|} \sum_{\tilde{a} \in \mathcal{A}(s)} V_{\phi_{i+1}}(s \circ \tilde{a})$$

25:       Add the estimated advantages into $\mathcal{D}_{\text{adv}}$, i.e.,

$$\mathcal{D}_{\text{adv}} \leftarrow \mathcal{D}_{\text{adv}} + \left\{ \left( s, a, \hat{A}(s, a) \right) : a \in \mathcal{A}(s) \right\}$$

26:    **end for**

27:    **[Step B.2 Policy Optimization]**

28:    Update the actor (starting from $\theta_i$) by optimizing (12) on the advantage dataset $\mathcal{D}_{\text{adv}}$, i.e.,

$$\theta_{i+1} \leftarrow \arg\min_{\theta} \frac{1}{2} \mathbb{E}_{(s, a, \hat{A}) \sim \mathcal{D}_{\text{adv}}} \left[ \left( \frac{1}{\beta} \hat{A}(s, a) - \log \frac{\pi_\theta(a|s)}{\pi_{\text{ref}}(a|s)} \right)^2 \right]$$

29: **end for**

30: **Output:** Last iterate actor $\pi_{\theta_T}$, last iterate critic $V_{\phi_T}$.

