# OpenReview forum: "DAPO : Improving Multi-Step Reasoning Abilities of Large Language Models with Direct Advantage-Based Policy Optimization"
_NeurIPS.cc/2025/Conference — NeurIPS 2025 spotlight_

### Official Review · Reviewer_j5UX · 2025-07-01

**Clarity:** 3
**Significance:** 2
**Originality:** 3
**Rating:** 4
**Confidence:** 4

**Summary:**

This paper explores the role of reinforcement learning in enhancing the reasoning capabilities of large language models and highlights key challenges, particularly the issue of sparse rewards. Sparse rewards increase training variance in policy optimization and hinder accurate value function estimation in Actor-Critic methods. To address these limitations, the authors propose DAPO, a novel step-level offline RL algorithm with theoretical guarantees for improving LLM reasoning.

**Questions:**

* 1. This method is designed for discrete spaces, can it be extended to continuous data spaces?
* 2. Could you provide insights into how the coefficient of KL regularization $\beta$ affects the experimental results, since this is a critical parameter in offline RL?
* 3. Regarding the sampling process in Section~3.2, could you clarify (1) whether each trajectory $\tau_i$ contains only single states $s$, and (2) the precise meaning of $s_{T_i-1}$ in the Monte Carlo mean calculation?

**Ethical Concerns:**

["NO or VERY MINOR ethics concerns only"]

**Final Justification:**

The authors have addressed the majority of my concerns in a thoughtful and constructive manner, and I will maintain my score.

**Limitations:**

As is mentioned in the weakness section above.

**Quality:**

3

**Strengths And Weaknesses:**

**Strength:**

* The proposed DAPO algorithm introduces a principled step-level optimization approach using a pretrained critic, effectively addressing sparse reward and critic training challenges in LLM reasoning tasks.
* For the proposed method, the authors provide comprehensive proofs, ensuring the theoretical completeness and feasibility.
* Comprehensive experiments show consistent gains across mathematical and coding benchmarks.

**Weakness:**

* This method appears to require extensive adoption during the pre-training of the critic, placing excessively high demands on both resources and complexity.
* Additionally, since the adopted offline method and advantage estimation do not fully address the out-of-distribution problem in offline RL, this approach heavily relies on the coverage of the pre-training data.

---

> ### Author Rebuttal · Authors · 2025-07-30
>
> Thank you for reviewing our paper and providing helpful comments.
>
> **About the `Weakness` part**
> - Regarding the computational resource, we acknowledge that pre-training the critic is resource-intensive. A brief discussion of this limitation is explicitly included in Section E and lines 257–267. We also address potential approaches for more efficient critic pre-training in lines 628–633. Nevertheless, the computational investment is justified: As shown in Figure 3, DAPO consistently improves model performance throughout actor training. Table 3 further demonstrates that DAPO delivers superior performance gains over baseline methods even under comparable computational budgets (see discussions in lines 257–267).
> - Concerning reliance on data coverage, this is a common challenge in offline RL methods. Theoretical analyses of offline RL (e.g., [1]) frequently assume adequate data coverage. As DAPO is an offline method without online exploration, it reasonably requires sufficient data coverage to achieve performance gains. Importantly, our experiments show that merely using samples generated by the reference model as the empirical training distribution suffices to yield measurable improvements.
>
> [1] Kishan Panaganti, Zaiyan Xu, Dileep Kalathil and Mohammad Ghavamzadeh. Bridging Distributionally Robust Learning and Offline RL: An Approach to Mitigate Distribution Shift and Partial Data Coverage. Arxiv.2310.18434
>
> **About the `Questions` part**
> - Our method can be extended to MDPs with continuous state and action spaces, as the Performance Difference Lemma (see line 96) and the definition of visit probability (see line 98) can be generalized to the continuous setting. However, handling continuous data spaces requires additional technical details. To avoid technical complications, we focus solely on the discrete setting, which also aligns with the data space of LLMs.
> - The results of our ablation study on the KL coefficient $\beta$ are detailed in lines 249–253. Generally, a smaller $\beta$ yields a more conservative policy learning and lower performance gains. Conversely, a larger $\beta$ enables more aggressive policy learning (potentially improving performance) but risks higher distribution shift, demanding greater data coverage. Thus, a moderate $\beta$ achieves the optimal trade-off.
> - (1) Each trajectory may contain multiple states. For example, denoting a trajectory as $\tau = (x, a_0, \ldots, a_{T-1})$, where $x$ is the prompt, $a_i$ is the token generated at index $i$ and $T$ is the response length. We
>  use the line break$\backslash$n to segment the response $y=(a_0,...,a_{T-1})$ into multiple steps. Assuming there are $k$ steps in total, the token sequence of the trajectory $\tau$ truncated at each $i$-th step can be regarded as a state. Thus, this process yields $k$ states in total.  (2) Each state $s$ (a context prefix) serves as the starting point for sampling $N$ sub-trajectories $(\hat{y}_1, \ldots, \hat{y}_N)$. We then compute the group mean reward  of these sub-trajectories as the critic training label of input state $s$. This procedure explains the Monte Carlo mean calculation mentioned.

---

> ### Comment · Reviewer_j5UX · 2025-08-01
>
> Thank you for your response. Most of my concerns have been addressed, and I believe this is a strong paper. I'm confident it can make a positive contribution to the LLM community.

---

### Official Review · Reviewer_xRdV · 2025-07-01

**Clarity:** 3
**Significance:** 3
**Originality:** 2
**Rating:** 4
**Confidence:** 3

**Summary:**

Paper "DAPO: Improving Multi-Step Reasoning Abilities of Large Language Models with Direct Advantage-Based Policy Optimization" proposes a novel policy optimization method for multi-step reasoning. It proposes dense step-level training signals by computing an advantage function at each intermediate step using a separately trained critic. Experiments on mathematical and code benchmarks show that DAPO consistently improves performance across multiple models (SFT and RL-pretrained), outperforming both PPO and GRPO baselines.

**Questions:**

1. The DAPO objective seems pretty interesting and novel. I saw similar squared loss in the IPO paper [1]. Could the authors discuss the relation of the proposed work and this reference?
2. Equation (9) should plus the gradient since we are doing gradient ascent? Also for Lemma 3.1 and (10), the signs should be positive?
3. Is the multi-step capability of the proposed framework mainly comes from the critic model V? That is, V is serving as the intermediate signal of the multi step states and actions? If so, I'm curious if the cross entropy training of V in line 169 (which is not new) could be improved since the current loss is still only making use of the outcome reward?
4. It is really astonishing to see the severe overfitting of GRPO in Fig. 4 for MATH dataset. Could the author comment on why DAPO could avoid such overfitting. Also would PPO and other algorithm also result in similar overfitting?
5. The paper uses dense RL notation that may not be easily digestible for a mixed NLP/ML audience. For example, input prompt could be denote as x and output continuation as y; while input, or input truncated with unfinished continuation could be denoted as state s, and the next chunk of continuation could be denoted as action a. I think this could lead to less confusion.
6. Typos: Line 190 "... reproducibility and of DAPO..."

**Ethical Concerns:**

["NO or VERY MINOR ethics concerns only"]

**Limitations:**

See questions section.

No potential negative societal impact.

**Quality:**

3

**Strengths And Weaknesses:**

Strengths: The work proposes a well-motivated solution using step-level optimization and separate critic training. The new surrogate loss could still lead to a monotonic improvement theorem (Theorem 3.2). DAPO consistently improves performance across in-domain (MATH) and out-of-domain benchmarks (GSM8K, Minerva, Olympiad, etc.).

Weaknesses (Please respond directly to the questions section): Training the critic is expensive due to massive rollout sampling. The accuracy of advantage estimates relies entirely on the quality of the critic.

---

> ### Author Rebuttal · Authors · 2025-07-30
>
> Thank you for reviewing our paper with helpful comments. Below we address each of the points you raised in `Questions` section one by one.
>
> - The squared loss studied in IPO is
> \begin{align}
> \min_{\theta} E_{\left(x,y_{w},y_{l} \right) \sim \mathcal{D}} \left[ \left( \log \left( \frac{\pi_{\theta} \left( y_{w}|x \right)}{\pi_{\text{ref}} \left( y_{w}|x \right)} \right) -\log \left( \frac{\pi_{\theta} \left( y_{l}|x \right)}{\pi_{\text{} \text{ref}} \left( y_{l}|x \right)} \right) -\frac{1}{2\beta} \right)^{2} \right],
> \end{align}
>     while the squared loss studied in DAPO is
>     $$
>     \underset{\theta}{\min} \frac{1}{2}E_{s\sim \nu _{\mathcal{S}},a\sim \nu _{\mathcal{A}}\left( \cdot |s \right)}\left[ \left( \frac{1}{\beta}A^{\pi _{\mathrm{ref}}}\left( s,a \right) -\log \frac{\pi _{\theta}\left( a|s \right)}{\pi _{\mathrm{ref}}\left( a|s \right)} \right) ^2 \right],
>     $$
>     where $\beta$ is the coefficient of KL regularization. **Although both objectives utilize the squared loss, they exhibit several fundamental differences.** For instance: 1) IPO is a preference learning algorithm requiring pairwise data, while DAPO is not. 2) IPO is a response-level algorithm as it optimizes the response probability directly, whereas DAPO is a step-level algorithm optimizing only the next-step distribution. However, the shared application of squared loss enables both algorithms to modulate the degree of policy adaptation through the KL regularization term $\beta$. Generally, a smaller $\beta$ yields a more conservative learning approach with lower performance gains. Conversely, a larger $\beta$ facilitates more significant policy updates (potentially improving performance) but increases the risk of distribution shift, demanding broader data coverage.
>
> - Thanks for pointing out the typo. The sign should be positive in Equation (9). We will fix it in the revised revision. The negative sign in (10) is correct since Lemma 3.1 shows that $\nabla_\theta \mathcal{I}_s(\pi_{\theta_k},\pi_{\text{ref}})=-\beta \nabla_\theta\mathcal{H}^k_s(\theta_k)$.
>
> - **The multi-step capability originates from the critic model $V$, necessitating critic pretraining in DAPO to ensure $V$ provides accurate intermediate signals for policy optimization**. Note that we only consider the binary outcome reward in this work and the value function defined in Equation (3) is a probability. Thus we use the binary cross-entropy loss in critic pretraining. Given the value function's definition (Equation (3)), we find it challenging to make fundamental improvements to critic pretraining merely through training loss design. Note that the critic pretraining objective (see Equation (14)) involves two elements: (1) the training state distribution $\mathcal{D}$ and (2) the state-wise loss function $\mathcal{L}$. For more efficient critic pretraining, we suggest further investigation into designing $\mathcal{D}$. Related discussion also appears in lines 628-633.
>
> -   **We attribute the overfitting of GRPO observed in Figure 4 primarily to the high training variance of response-level advantage estimation.** As discussed in Section D, some responses yield correct answers despite containing flawed reasoning steps that may lead the model to make mistakes. Consequently, using response-level advantage estimation in RL training can overfit to such responses and harm OOD performance. Our PPO experiment on Qwen2.5-Math-7B-Instruct exhibits similar overfitting. We find that the standard error of the critic output decays to zero. This indicates that given response $y$, the state value predictions for tokens in $y$ output by the critic model converge to some constant $c$, causing the critic to fail in providing meaningful intermediate signals. In this scenario, the GAE($\lambda$) estimator used in PPO satisfies $A_t \rightarrow \lambda^{T-t-1}(r(x,y)-c)$, effectively reducing PPO to a response-level algorithm like GRPO. DAPO pretraining the critic using Section 3.2's procedure ensures accurate intermediate signals for policy optimization, yielding more robust performance improvements.
> -  Thanks for your helpful advice.
> - Thanks for pointing out this typo; we'll fix it.

---

> > ### Comment · Reviewer_xRdV · 2025-08-05
> >
> > Thank you for your detailed response. Your clarifications addressed my main concerns and I'll maintain my positive score.

---

### Official Review · Reviewer_SP6p · 2025-07-02

**Clarity:** 2
**Significance:** 2
**Originality:** 2
**Rating:** 4
**Confidence:** 2

**Summary:**

The paper introduces DAPO, a step-level offline RL method designed to enhance the multi-step reasoning capabilities of LLM in tasks like mathematical problem-solving and code generation. Unlike response-level RL methods that rely on sparse final rewards, DAPO addresses high training variance by using a critic function to provide dense, step-level training signals. The method trains the critic independently before optimizing the policy, avoiding the instability of traditional Actor-Critic methods. Experiments on mulyiple benchmarks demonstrate DAPO's superior performance.

**Questions:**

1. The paper mentions Step-DPO but does not include a direct performance comparison in the experiments. Could you add a comparison with Step-DPO to highlight DAPO's superiority?

2. DAPO separates the training of the actor and critic into two stages to improve stability. Could this lead to the critic being unable to adapt to changes in the actor's policy? How does DAPO solve this problem?

**Ethical Concerns:**

["NO or VERY MINOR ethics concerns only"]

**Final Justification:**

The authors’ rebuttal provided clear responses to my main concerns, especially regarding the separation of actor and critic training and the rationale for not including step-DPO as a baseline. While issues of data coverage and computational requirements remain, these are common limitations in offline RL. The theoretical contributions and empirical results are solid. Considering the clarifications, I have raised my score.

**Quality:**

3

**Strengths And Weaknesses:**

Strength

1. DAPO introduces a step-level offline reinforcement learning (RL) approach, addressing the sparse reward problem in multi-step reasoning tasks.
2. The paper provides a rigorous theoretical foundation,  proving monotonic policy improvement under KL-regularized objectives
3. Extensive experiments demonstrate DAPO’s effectiveness across multiple mathematical and coding benchmarks

Weakness

1. DAPO's performance heavily relies on the quality and diversity of the training dataset. The paper does not sufficiently discuss how to handle cases where the dataset lacks coverage for certain states or actions, which is a common challenge in offline RL.
2. Generating multiple sub-trajectories for critic training and sampling multiple actions for advantage estimation may introduce significant computational costs. For researchers or institutions with limited resources, generating high-quality training states and sub-trajectories could be a bottleneck, especially for larger-scale models or more complex tasks.
3. While DAPO compares favorably to PPO and GRPO, it does not benchmark against other recent offline RL methods, such as step-DPO [1].
4. DAPO's experiments primarily focus on mathematical reasoning and code generation, which have clear correctness rewards, but the paper does not validate DAPO's effectiveness in other tasks (e.g., open-domain question answering).

[1]Step-DPO: Step-wise Preference Optimization for Long-chain Reasoning of LLMs

---

> ### Author Rebuttal · Authors · 2025-07-31
>
> Thank you for reviewing our paper with helpful comments! We address your concerns point by point below.
>
> Regarding the `Weaknesses` section:
>
> - We acknowledge that the performance of DAPO relies on data coverage, as it is a typical offline RL method and lacks online exploration. However, as you pointed out, this is a common challenge in offline RL. Theoretical analyses of offline RL also frequently assume adequate data coverage. Thus, addressing the data coverage problem thoroughly is beyond the scope of this work. That being said, our experiments show that merely using trajectories generated by the reference model as the empirical training distribution suffices to yield measurable improvements. This suggests that data coverage may not pose a severe problem for achieving performance improvements in the trust region of reference policy.
>
> - Regarding the computational resource demands raised as a weakness, we acknowledge that pretraining the critic is resource-intensive. A brief discussion of this limitation is already explicitly included in Section E and lines 257–267. We also address potential approaches to reduce the computation cost for critic pretraining in lines 628–633. **It should be noted, however, that achieving higher performance improvements generally requires generating more samples and consequently demands greater computational resources. Our DAPO provides a valid method for attaining such improvements when adequate training resources are available**. As shown in Figure 3, DAPO consistently enhances model performance throughout actor training. Table 3 further demonstrates that DAPO delivers superior performance gains over baseline methods **even under comparable computational budgets**(see discussion in lines 257–267).
>
> - Note that Step-DPO is a preference learning algorithm that requires the pairwise preference dataset $\mathcal{D}$, and the curation of $\mathcal{D}$ dominates its performance. **Consequently, conducting a fair performance comparison between DAPO and Step-DPO is challenging**. Therefore, we chose PPO and GRPO, instead of Step-DPO, as our baseline methods.
>
> - For RL training on tasks lacking clear correctness rewards, a reward model is often utilized. **However, reward hacking frequently occurs in such scenarios. To prevent interference from reward hacking when using reward models and to validate the effectiveness of the RL algorithm itself, we focus DAPO's experiments on mathematical reasoning and code generation**. These tasks are verifiable and provide intrinsic correctness rewards, thus minimizing the likelihood of reward hacking. Furthermore, as theoretically guaranteed (see Theorem 3.2), it is reasonable to anticipate that DAPO will achieve performance improvements on the training distribution, even for tasks without clear correctness rewards.
>
> Regarding the `Questions` section:
>
> - The reason why we do not include step-DPO for performance comparison is mentioned above. **The main superiority of DAPO over step-DAPO  is that DAPO eliminates the need to curate a high-quality preference dataset to provide optimization signals for intermediate steps**, as producing such a dataset is often challenging or equally computationally expensive. Our experiments demonstrate that performing simple Monte Carlo sampling from the reference model achieves measurable improvements. Furthermore, for a given input context prefix, DAPO can learn directly from multiple samples, whereas step-DPO requires pairwise data.
>
> - While actor-critic methods often update the actor and critic simultaneously, our core theoretical result (Theorem 3.2) shows that DAPO's policy optimization against the **fixed** value function of the reference model guarantees performance improvement. **Therefore, the critic's inability to adapt to the actor's policy changes within a single iteration is not problematic**. It is worth noting that we also conducted experiments with iterative DAPO. In this approach, the actor model optimized by DAPO becomes the new $\pi_\text{ref}$, and the critic is updated to estimate the value function of this new reference model for further policy optimization in the next iteration. The results (lines 243-248) show that adapting the critic to the actor's changes leads to greater performance gains.
>
> **We sincerely hope our responses have addressed your concerns and will positively influence your assessment of our paper.**

---

> > ### Comment · Reviewer_SP6p · 2025-08-01
> >
> > Thank you for your detailed response. Your clarifications addressed my main concerns, so I have decided to increase my score.

---

### Official Review · Reviewer_8sNP · 2025-07-04

**Clarity:** 3
**Significance:** 3
**Originality:** 3
**Rating:** 4
**Confidence:** 3

**Summary:**

This paper introduces Direct Advantage-Based Policy Optimization (DAPO), a new reinforcement learning algorithm designed to improve the reasoning capabilities of large language models (LLMs). DAPO addresses two key challenges in applying RL to LLM reasoning: high training variance in response-level algorithms and poor value estimation in actor-critic methods. The method works by first training a critic function independently to estimate state values, then using these estimates to compute advantages for optimizing the policy at each reasoning step. Unlike existing response-level methods that apply uniform updates across all steps based on final rewards, DAPO provides step-specific training signals through the critic. The authors validate DAPO on mathematical and coding tasks, demonstrating consistent improvements over baseline methods like GRPO and PPO across multiple model architectures. The paper provides theoretical guarantees showing that DAPO produces better policies until reaching an optimal reference policy, along with detailed empirical analysis of its computational requirements and implementation considerations.

**Questions:**

- What is the step by step algorithm of DAPO?
- Can you show ablation results with MSE loss for the critic as is typically used in literature?
- What is the stopping criteria for the DAPO training process?
- How much is the computational cost in terms of training time, FLOPs? Which machines were used?
- It has been mentioned that "only 4K samples" are used in training? How small is that dataset compared to related works?
- Why does performance drop in College Math and LiveCodeBench?

**Ethical Concerns:**

["NO or VERY MINOR ethics concerns only"]

**Final Justification:**

Author responses did not change my assessment

**Limitations:**

Limited discussion of failure cases or scenarios where the method might not work well

**Quality:**

3

**Strengths And Weaknesses:**

Strengths:
- Strong theoretical foundation with clear proofs and guarantees for policy improvement
- Novel approach to addressing sparse rewards by using step-level advantages instead of response-level signals
- Comprehensive empirical validation across multiple model architectures and benchmarks
- Clear improvement over baseline methods (GRPO and PPO) shown in Tables 1-3
- Ablation studies on hyperparameters and implementation considerations

Weaknesses:
- Critic training requires generation from base models and segmenting the generated text. It is unclear if this step is generalizable across models and (reasoning) problems.
- No analysis of performance with weaker base models - all experiments use strong foundation models that are already trained with RL
- Computational cost analysis is buried in Appendix E and lacks clear explanation of practical implementation requirements
- Missing step-by-step algorithm description for implementing DAPO in practice
- Empirical results in Figure 3 show fluctuations in performance, contradicting theoretical claims of monotonic improvement
- Nit: Poor paper organization with key figures (like Figure 3) referenced in main text but placed in appendix
- Nit: Technical terms like "cot" and "PRM" are not defined

---

> ### Author Rebuttal · Authors · 2025-07-31
>
> Thank you for reviewing our paper and providing helpful comments. Let us address your concerns in the following.
>
> Regarding the `Weakness` section:
>
> - From our perspective, this step can be easily extended to other models. The potential obstacle preventing its generalization to other tasks may lie in how to segment the generated text into distinct steps. In our work, we use the newline character $\backslash$n to segment the text. This method is not necessarily optimal for long chain-of-thought scenarios or other tasks (such as agentic RL). We have discussed this breifly in lines 628-633.
>
> - Our experiments were **not** conducted exclusively on stronger models after RL training. As shown in Table 1, we tested our method on four SFT models and three RL models, covering both stronger and weaker variants. For the weaker SFT model like Skywork-Math-Llama, our algorithm was applied and achieved performance improvement. Table 4 further demonstrates that iterative DAPO can further enhance this model’s performance optimized by DAPO. Our approach also yielded notable improvements on stronger RL models, such as Qwen2.5-Math-7B-Instruct, demonstrating the effectiveness of DAPO.
>
> - Thanks for providing helpful feedback regarding the computational cost analysis. We would like to include a detailed analysis of training FLOPs of DAPO in our experiments in revised version.
>
> - Thanks for your feedback regarding the algorithmic description of DAPO. While the current implementation steps are outlined in Figure 1 and Lines 205–216, we acknowledge that a more granular, step-by-step specification would strengthen practical reproducibility. To address this, we will add a detailed pseudocode description of DAPO in the appendix of the revised version.
>
> - The performance fluctuations observed in Figure 3 **do not contradict** our theoretical claim in Theorem 2. We wish to clarify that the monotonic improvement property (established in Theorem 3.2) specifically compares the converged DAPO-optimized policy against the reference policy --- not adjacent gradient steps in the training stage. As evidenced in Figure 3, all  policies after DAPO training with adaquate gradient steps consistently outperform the initial reference policy. This alignment with theory holds true across all tested models. The transient fluctuations within the training process are attributable to stochastic training data.
>
> - We appreciate your helpful suggestions regarding the paper's writing and will make appropriate adjustments accordingly.
>
> Regarding the `Question` section:
>
> - The step-by-step algorithm is already detailed in Figure 1 and Lines 205–216. Briefly: during critic pretraining, we first use the base model to generate multiple solutions for each training problem. Each selected solution is segmented into states (using $\backslash$n as the delimiter in this work). For every state $s$ (intermediate step), we perform multiple completions using the base model as the completer. We then construct Monte Carlo mean rewards as training labels for $s$ and optimize the critic network using BCE loss (Line 169). After critic pretraining, we extract next-steps (actions) for each state by segmenting completions into individual steps. Advantages are computed using the pretrained critic (Lines 152–158). This advantage dataset then optimizes the model through DAPO loss (Equation 12; Lines 149–151). **For clearer algorithmic specification, pseudocode for DAPO will be provided in the appendix of the revised manuscript.**
>
> - As stated in line 170, we use BCE loss to accelerate the critic's training. While we did not empirically compare MSE and BCE losses, our choice of BCE loss was theoretically motivated: Consider the critic objective in Equation (14). The gradients of sample-wise loss functions are given by
> \begin{align*}
> \nabla_\phi \mathcal{L}\{MSE}\left(V\phi(s),V^\pi(s)\right) &= \left(V^\pi(s)-V_\phi(s)\right)\nabla_\phi V_\phi(s)
> \end{align*}and
> \begin{align*}
> \nabla_\phi \mathcal{L}\text{BCE}\left(V\phi(s),V^\pi(s)\right) &= \dfrac{\left(V^\pi(s)-V_\phi(s)\right)}{V_\phi(s)\left(1-V_\phi(s)\right)}\nabla_\phi V_\phi(s).
> \end{align*}
> This shows that the BCE loss gradient is equivalent to the MSE loss gradient with an adaptive sample-wise learning rate $\frac{1}{V_\phi(s)(1-V_\phi(s))} \ge 4$. This accelerates critic training, particularly when $V_\phi(s)\rightarrow 0$ or $1$.
>
> - We stop the DAPO training process when the batch mean of absolute log ratio, i.e. $\hat{E}_{\left( s,a \right)} \left[ \left| \log \frac{\pi_{\theta} \left( a|s \right)}{\pi_{\text{ref}} \left( a|s \right)} \right| \right]$, converges, indicating policy convergence.
>
> - We use A100 GPUs for both generation and training. As discussed in Section E, the total FLOPs consumed during the entire training stage can be approximated by $2N_\text{model}(N_\text{inference} + 3N_\text{training})$. In our DAPO experiment on Llama-3.1-8B-Instruct, $N_\text{inference} \approx 1.65 \times 10^9$ and $N_\text{training} \approx 9.68 \times 10^7$. This total computational cost is roughly equivalent to 1k steps of online RL training in GRPO with a batch size of 1024 and a maximum of 2048 new tokens
>
> - The TACO dataset originally contains 26,443 questions. After filtering, only 4k questions were retained for RL training, which is much smaller than the original dataset. This is why we emphasize that there are only 4k samples.
>
> - The performance drop in College Math and LiveCodeBench may be due to limited training samples which lack enough diversity.

---

> > ### Comment · Reviewer_8sNP · 2025-08-05
> >
> > Thank you for the responses. Most of my questions have been observed.
> >
> > I still think the generalization of n-steps separation method is important for this algorithm to be applied to arbitrary problems. It is unclear if this can be achieved from the evidence provided.
> >
> > However, the paper still makes significant contributions, and I'm happy to keep my current score.

---

### Decision · Program_Chairs · 2025-09-17

**Decision:**

Accept (spotlight)

**Comment:**

The reviewers found DAPO to be a technically solid and well-motivated contribution. They consistently highlighted the novelty of introducing step-level optimization with an independently trained critic, strong theoretical guarantees of monotonic policy improvement, and compelling empirical validation across multiple mathematical and coding benchmarks. The method was seen as an effective way to address the sparse reward and high-variance issues in response-level RL, with clear performance improvements over PPO and GRPO. The authors’ rebuttal successfully clarified questions about critic training, computational costs, and connections to related methods (e.g., Step-DPO, IPO), and most reviewers expressed satisfaction that their main concerns were addressed.

The main points of disagreement centered on practical limitations: the heavy reliance on data coverage, high computational cost of critic training, and uncertainties about generalization to tasks beyond math/code reasoning. Some reviewers also noted missing baselines and the lack of broader task validation. However, these issues were considered common challenges in offline RL rather than fatal flaws, and did not outweigh the paper’s contributions. Overall, I find this an insightful and well-executed paper, and support its acceptance and NeurIPS.